



# Biological and biogeochemical methods for estimating bio-irrigation: a case study in the Oosterschelde estuary.

Emil De Borger[1,2], Justin Tiano[2,1], Ulrike Braeckman[1], Tom Ysebaert[2,3], Karline Soetaert[2,1].

[1]Ghent University, Department of Biology, Marine Biology Research Group, Krijgslaan 281/S8, 9000 Ghent, Belgium
[2]Royal Netherlands Institute of Sea Research (NIOZ), Department of Estuarine and Delta Systems, and Utrecht University, Korringaweg 7, P.O. Box 140, 4401 NT Yerseke, The Netherlands
[3]Wageningen Marine Research, Wageningen University & Research, Wageningen, Netherlands

*Correspondence to* : Emil De Borger (emil.de.borger@nioz.nl)

**Abstract**

Bio-irrigation, the exchange of solutes between overlying water and sediment by benthic organisms, plays an important role in sediment biogeochemistry. Quantification of bio-irrigation is done either through measurements with tracers, or more recently, using biological traits to derive the community (bio-) irrigation potential (IPc). Both these techniques were applied in a seasonal study of bio-irrigation in species communities of subtidal and intertidal habitats in a temperate estuary. A tracer time series with high-temporal resolution allowed to simultaneously estimate the pumping rate, and the sediment attenuation, 15   a parameter that determines irrigation depth. We show that although the total pumping rate is similar in both intertidal and subtidal areas, there is deeper bio-irrigation in intertidal areas. This is explained by higher densities of bio-irrigators such as *Corophium sp.*, *Heteromastus filiformis* and *Arenicola marina* in the intertidal, as opposed to the subtidal. The IPc correlated more strongly with the attenuation coefficient than the pumping rate, which highlights that this index reflects more the bio-irrigation depth rather than the rate.

**1   Introduction**

Bio-irrigation is the process in which benthic organisms actively or passively exchange sediment porewater solutes with the overlying water column as a result of burrowing, pumping, ventilation and feeding activities (Kristensen et al., 2012). This exchange plays an important role in marine and lacustrine sediment biogeochemistry, as oxygen rich water is brought into an otherwise sub- or anoxic sediment matrix. This allows for aerobic degradation processes to take place, as well as the reoxidation 25   of reduced substances (Aller and Aller, 1998; Kristensen, 2001), and enables sediment dwelling organisms to forage and live in the otherwise anoxic deeper sediment layers (Olaffson, 2003; Braeckman et al., 2011). Bio-irrigation also enhances nutrient exchange by extending the exchange surface (Quintana et al., 2007), and increasing degradation rates (Na et al., 2008). Sedimentary bio-irrigation is the result of the combined actions of a multitude of organisms sharing the same habitat. Some organisms such as the smaller meiofauna, located close to the sediment water interface, exchange only small amounts of



solutes, but due to their high densities they affect the sediment porosity and as such exert a significant effect on sediment water exchanges in the top layers of the sediment (Aller and Aller, 1992; Rysgaard et al., 2000). On the opposite end of the spectrum are larger infaunal species such as the burrowing shrimp *Upogebia pugettensis* (Dana, 1852) which constructs burrows that extend up to 1 m into the sediment and that actively ventilate these burrows using their pleiopods (D'Andrea and DeWitt, 2009). These deep burrows substantially extend the oxic sediment-water interface into the sediment, influencing the associated

microbial respiration through various pathways (Nielsen et al., 2004). Bio-irrigation also depends on the sediment matrix. In muddy sediments bio-irrigation impacts are localized close to the burrow wall, as the transport of solutes radiating from the burrows is governed by diffusion (Aller, 1980). In sandy, more permeable sediments the pressure gradients caused by pumping activities induce water flows through the surrounding sediments, thus affecting the sediment matrix further away from the burrow walls (Meysman et al., 2005; Timmermann et al., 2007). Therefore, the effects of bio-irrigation depend on a

combination of the species community, species' individual behavior including pumping activity, the depths at which they occur, and the sediment matrix they inhabit.

Bio-irrigation can be quantified with biogeochemical methods, or estimated using an index of bio-irrigation based on biological information. The biogeochemical methods measure the exchange of a tracer substance (usually inert) between the overlying water and the sediment. Bio-irrigation rates are then either estimated from the tracer concentration time series in the overlying

water, from the distribution profile of the tracer in the sediment at the end of the incubation, or using both. The fit of the tracer concentration is done using either a quasi-mechanistic model (Berelson et al., 1998; Na et al., 2008; Andersson et al., 2006), or by applying a linear regression (Mestdagh et al., 2018; De Smet et al., 2016; Wrede et al., 2018). The bio-irrigation estimates obtained by these methods have the advantage that they are easily applied in mathematical models that represent the sediment biogeochemistry.

The index approach starts with the quantification of the abundance and biomass of organisms in the sediment, and an assessment of how the different species might contribute to bio-irrigation based on a set of life history traits: burrow type, feeding type and burrowing depth. The species biomass and abundance, combined with their trait scores are then used to derive an index that represents the community (bio-)irrigation potential (BIPc and IPc in Renz et al., 2018 and Wrede et al., 2018 respectively), a similar practice to what is done for bioturbation with the community bioturbation potential (BPc; Queirós et

al., 2013). The advantage of biologically-based indices is that large datasets of benthic communities are currently available (e.g. Craeymeersch et al., 1986; Degraer et al., 2006; Northeast Fisheries Science Center, 2018), so that these data have great potential to derive information on the temporal and spatial variability of bio-irrigation. Unfortunately, in contrast to the related bioturbation potential (Solan et al., 2004), the classification of sediments according to their bio-irrigation potential is a very recent endeavor, and the underlying mechanistic basis of these indices, i.e. what they actually describe, should be explored

further. One notable exception is the IPc index of Wrede et al. (2018) which has been calibrated against bromide uptake rates for selected individual species and communities in the German Bight of the North Sea.

To assess the congruity of both types of bio-irrigation measurements along an environmental gradient, we measured bio-irrigation rates and bio-irrigation potential in a temperate estuarine system, the Oosterschelde. Three subtidal and three





intertidal sites with different benthic communities, and sediments varying from muddy to sandy, were sampled in across
different seasons. Bio-irrigation rates were derived by fitting a mechanistic model through a quasi-continuous time series of a
fluorescent tracer (Meysman et al., 2006; Na et al., 2008), while biological information was used to estimate the IPc and BPc
indices.

## 2 Materials and methods

### 2.1  Sampling

Field cores were collected in the Oosterschelde (SW Netherlands) from August 2016 to December 2017 (Fig. 1). Six sites (3
subtidal, 3 intertidal) were selected based on results from previous sampling efforts, to reflect the variability in inundation time
and sediment composition present in this area (Table 1). The intertidal sites Zandkreek (N 51.55354°, E 3.87278°), Dortsman
(N 51.56804°, E 4.01425°) and Olzendenpolder (N 51.46694°, E 4.072694°) were sampled by pressing two cylindrical PVC
cores (14.5 cm Ø, 30 cm height) into the sediment at low tide up to a depth of 20 cm at most, and extracting them from the
sediment. The subtidal sites Hammen (N 51.65607°, E 3.858717°), Viane (N 51.60675°, E 3.98501°), and Lodijksegat (N
51.48463°, E 4.166001°) were sampled in the same way, but sediment was retrieved from duplicate deployments of a NIOZ
box-corer aboard the Research Vessel Delta. In total 70 samples in the intertidal, and 47 in the subtidal were retrieved. At each
site additional samples for sediment characteristics (sediment grain size and porosity) and chlorophyll $a$ content were collected
with a cut-off syringe retrieving the top 2 cm of the sediment.
After transportation to the laboratory, the cores were placed into buffering seawater tanks in a climate room set to the average
water temperature of the month in which the samples were taken. By adding 0.45 µm filtered Oosterschelde water, the
overlying water height was brought to at least 10 cm, and air stones and a stirring lid (central magnet stirrer) with sampling
ports were used to keep the water oxygenated. The sediment cores were left to acclimatize for 24 to 48 hours before starting
the irrigation experiment. For the irrigation measurements, a stock solution of 1 mg L$^{-1}$ uranine (sodium fluoresceine -
$C_{20}H_{10}NaO_5^-$) was prepared by dissolving 1 mg of uranine salts into 1 L of .45 µm filtered Oosterschelde water. To start the
experiment 30 to 40 mL of the stock solution was added to the overlying water to achieve a starting concentration of uranine
of about 10 µg L$^{-1}$. The concentration of the fluorescent tracer was subsequently measured every 30 seconds for a period of at
least 12 hours with a fluorometer (Turner designs cyclops 6) placed in the water column through a sampling port in the stirring
lid of the core, ± 6 cm below the water surface. After the measurement, the sediment was sieved over a 1 mm sieve and the
macrofauna was collected and stored in 4% buffered formalin for species identification and abundance and biomass
determination.
Sediment grain size was determined by laser diffraction on freeze dried and sieved (< 1 mm) sediment samples in a Malvern
Mastersizer 2000 (McCave et al., 1986). Water content was determined as the volume of water removed by freeze drying wet
sediment samples. Sediment porosity was determined from water content and solid phase density measurements, accounting
for the salt content of the pore water. Chl $a$ was extracted from the freeze dried sediment sample using acetone, and quantified





through UV spectrophotometry (Ritchie, 2006). The C/N ratio was calculated from total C and N concentrations, determined using an Interscience Flash 2000 organic element analyser.

### 2.2 Model

The exchange of a tracer ($T$) between the sediment and the overlying water is described in a (vertical) one-dimensional mechanistic model, that includes molecular diffusion, adsorption to sediment particles, and bio-irrigation. The bio-irrigation is implemented as a non-local exchange in which a pumping rate ($r$) exponentially decays with distance from the surface (z). This exponential decay mimics the depth dependent distribution of faunal biomass often found in sediments (Morys et al., 2017) and the associated decreasing amount of burrow cross-sections with depth (Martin and Banta, 1992; Furukawa et al., 2001).

The mass balance for a dissolved tracer ($T$, Eq. 1): and the adsorbed tracer ($A$, Eq. 2) in an incubated sediment with height $h_s$, at a given depth (z, cm) and time (t, hours) in the sediment is:

$$\frac{\partial T_z}{\partial t} = \frac{1}{\varphi_z} \cdot \frac{\partial}{\partial z}\left[D_s \varphi_z \frac{\partial T_z}{\partial z}\right] + r \frac{e^{-az}}{\int_0^{hS} e^{-az} dz} \cdot (T_{OW}- T_z) - k \cdot (Eq_A \cdot T_z - A_z) \cdot \rho \cdot \frac{(1-\varphi_z)}{\varphi_z} \tag{1}$$

$$\frac{\partial A_z}{\partial t} = k \cdot (Eq_A \cdot T_z - A_z) \tag{2}$$

In this equation $\varphi_z$ is sediment porosity, and $\rho$ is sediment density.

In the equation for $T$ (Eq. 1), the first term represents transport due to molecular diffusion, where $Ds$ is the sediment diffusion coefficient (cm² h⁻¹). The second term represents the exchange of tracer between the water column ($T_{OW}$) and any sediment depth $z$ due to irrigation, where the exchange rate decreases exponentially as modulated by the attenuation coefficient $a$ (cm⁻¹). The exponential term is scaled with the integrated value, so that the exchange rate $r$ reflects the total rate of bio-irrigation, expressed in (cm h⁻¹).

The loss term for the tracer by adsorption (third term) depends on the deviation from the local equilibrium of the tracer with the actual adsorbed fraction on the sediment and with parameters $k$ (h⁻¹), the rate of adsorption, and $Eq_A$, the adsorption equilibrium (ml g⁻¹).

The dissolved tracer concentration in the water column ($T_{OW}$) (Eq. 3) decreases by the diffusive flux into the sediment and the integrated irrigation flux, corrected for the thickness of the overlying water ($h_{OW}$):

$$\frac{\partial T_{OW}}{\partial t} = \frac{1}{h_{OW}}\left(-D_s \varphi_0 \frac{\partial T_z}{\partial z}\bigg|_{z=0} - \int_0^{hS} r \cdot \frac{e^{-az}}{\int_0^{hS} e^{-az} dz}(T_{OW}- T_z)dz\right) \tag{3}$$

The concentration of $A$ in the overlaying water equals 0.

The model was implemented in FORTRAN and integrated using the ode.1D solver from the R package deSolve (Soetaert et al., 2010; R Core Team, 2013). The sediment was subdivided into 50 layers; thickness of the first layer set equal to 0.5 mm and then exponentially increasing until the total sediment modelled was equal to the sediment height in each laboratory experiment.





### 2.3 Model fitting

Most of the input parameters of the model were constrained by physical measurements. Sediment porosity φ and specific density $\rho$ (g cm$^{-3}$) were derived from sediment samples taken alongside the cores in the field. The adsorption equilibrium $Eq_A$ (in ml g$^{-1}$) was determined from batch adsorption experiments (See supplementary data). The modelled sediment height ($h_S$) and water column height ($h_{OW}$) were set equal to the experimental conditions. This left two parameters governing the bio-irrigation rate to be estimated by model fitting: $r$, the integrated pumping rate and $a$, the attenuation coefficient. Fitting of the model to the experimental data was done with the R package FME (Soetaert and Petzoldt, 2010). First an identifiability analysis was performed to investigate the certainty with which these parameters could be derived from model fitting given the experimental data. This process entails a local sensitivity analysis (to quantify the relative effects of said parameters on model output), and a collinearity analysis (to test whether parameters were critically correlated, and thus not separately identifiable, or the opposite). Then both parameters were estimated by fitting the model to each individual tracer time series through minimization of the model cost (the weighted sum of squares) using the pseudo-random search algorithm (Price, 1977) followed by the Levenberg-Marquardt algorithm. Lastly, a sensitivity analysis was performed to calculate confidence bands around the model output, corresponding to the parameter covariance matrix derived from the fitting procedure.

### 2.4 Calculation of IP$_c$ and BP$_c$

The retrieved benthic macrofauna were identified down to lowest possible taxonomic level, counted and their ash-free dry weight (gAFDW m$^{-2}$) converted from blotted wet weight according to Sistermans et al. (2006). Based on the species abundance and biomass, the irrigation potential of the benthic community in a sediment core (IPc, Eq. 4) was calculated as described in Wrede et al. (2018):

$$IP_c = \sum_{i=1}^{n} \left(\frac{B_i}{A_i}\right)^{0.75} \cdot A_i \cdot BT_i \cdot FT_i \cdot ID_i \tag{4}$$

in which $B_i$ represents the biomass (gAFDW m$^{-2}$), $A_i$ the abundance (ind. m$^{-2}$) of species $i$ in the core, and $BT_i$, $FT_i$ and $ID_i$ are descriptive numerical scores for the species burrowing type, feeding type and injection pocket depth respectively. The values for $FT_i$, $BT_i$ and $ID_i$ were the same as applied by Wrede et al. (2018). If not available, values were assigned based on the closest taxonomic relative, with possible adjustments to correct for size differences and feeding type as taxonomic relation is not always a measure for similarity in traits.

The community bioturbation potential (BPc, Eq. 5) was calculated as described in Solan et al. (2004):

$$BP_c = \sum_{i=1}^{n} \left(\frac{B_i}{A_i}\right)^{0.5} \cdot A_i \cdot M_i \cdot R_i \tag{5}$$

With $M_i$ the mobility score and $R_i$ the reworking score for species $i$ from Queirós et al. (2013). Note that the biomass $B$ in this case is the blotted wet weight of the organisms.



### 2.5 Statistical analysis

Differences in model derived pumping rates $r$ and attenuation coefficient $a$ between subtidal and intertidal were tested using a two-sided T-test (assuming a significance level of 0.05). Since not all six stations were sampled on the same date, species densities, biomass, and estimated irrigation parameters were averaged per station, and per season (Table 2) for further multivariate analysis. The patterns in bio-irrigation rates were analysed using ordination techniques for multivariate datasets as described in Thioulouse et al. (2018), as implemented in the ade4 R package (Dray and Dufour, 2015). Absolute species abundances were transformed to relative abundances to standardize the analysis to species composition, and subjected to a centered principle component analysis (PCA). The continuous environmental variables were categorised to allow for performing a multiple correspondence analysis (MCA), which can account for possible nonlinear interactions between variables. Based on grain size, sediments were split into the Udden-Wentworth scale (Wentworth, 1922) of silt (< 63 µm), very fine sand (> 63 µm, < 125 µm) and fine sand (> 125 µm, < 250 µm); the Chl $a$ content was used to divide sites with low (< 8 µg g$^{-1}$), intermediate (8-16 µg g$^{-1}$) and high (> 16 µg g$^{-1}$) chlorophyll content. Together with the two other categorical variables habitat type (intertidal versus subtidal) and season, the environmental variables were subjected to MCA. Sediment porosity, and C/N ratio were not used in the analysis given the small range within these data (Table 2). The covariance of species and environmental datasets was then explored in a co-inertia analysis (CoiA). This is a stable method for data tables which contain multiple variables that could be correlated (Dray et al., 2003). In this symmetrical technique data tables with different ordination types are analysed simultaneously without assuming an explanatory-response relation, and eigenvalues (squared covariances between linear combinations of species abundances and environmental variables in the CoiA) are computed on the common structures of both datasets. The correlations between the response variables relating to irrigation (estimated irrigation parameters, calculated IPc, BP$_c$, and irrigation standardized per individual) and the two axes of the co-inertia analysis were then assessed using the Pearson correlation coefficient assuming a significance level of 0.05. Results are expressed as mean ± sd.

## 3 Results

### 3.1 Environmental variables

Sediment descriptors are summarized in Table 2. Chlorophyll a concentrations in the upper 2 cm of the sediment varied from 3.76 ± 2.43 µg g$^{-1}$ in Hammen to 20.60 ± 4.19 µg g$^{-1}$ in Zandkreek and were higher in the intertidal (13.34 ± 6.53 µg g$^{-1}$) than in the subtidal (5.88 ± 4.20 µg g$^{-1}$). In the intertidal, the median grain size (MGS) and silt content ranged from 59 µm with 52% silt to 140 µm with 0% silt. In the subtidal the range in grain size was broader, from 53 µm with 63% silt to 201 µm with 24% silt. The C/N ratio (mol mol$^{-1}$) was similar for all sites (9.3 ± 1.0 – 12.3 ± 1.4) with the exception of Dortsman, where values were lower (6.5 ± 1.2). Dortsman was also the site where the organic carbon content was lowest (0.07 ± 0.02 %), this value increased with silt content, to highest values in the most silty station Viane (1.16 ± 0.36 %).





### 3.2 Macrofauna

In total, 60 species were identified in the 6 different stations (Table 3). Species abundances in the intertidal were generally one, sometimes two orders of magnitude higher than in the subtidal (see Fig. 2: a, b for seasonal density and biomass data). In the intertidal, maximum abundances were observed in Dortsman in autumn and spring, with peak values of $15202 \pm 4863$ and

$16054 \pm 13939$ ind m$^{-2}$ respectively, mainly due to high abundances of the amphipods *Corophium sp.* and *Bathyporeia* sp. (respective peak values of $9957 \pm 4465$ and $3934 \pm 3087$ ind m$^{-2}$). Subtidal densities varied less and were highest in Lodijksegat in autumn and summer (peak values of $661 \pm 502$ and $790 \pm 678$ ind m$^{-2}$ respectively). Faunal biomass was larger in the subtidal ($22.31 \pm 26.42$ gAFDW m$^{-2}$) as opposed to the intertidal ($10.51 \pm 8.59$ gAFDW m$^{-2}$), with peak summer values at the subtidal Lodijksegat station ($39.90 \pm 34.87$ gAFDW m$^{-2}$) coinciding with high abundances ($972 \pm 172$ ind m$^{-2}$) of the common

slipper limpet *Crepidula fornicata* (Linnaeus, 1758).

### 3.3 Bio-irrigation rates

A typical time series of uranine concentrations shows the tracer to exponentially decrease towards a steady value (Fig. 3a). The pumping rate and irrigation attenuation (parameters $r$ and $a$) have an opposite effect on tracer concentrations in the overlying water, but a collinearity analysis (Soetaert and Petzoldt, 2010) showed that these two parameters could be fitted

simultaneously. The attenuation coefficient $a$, affects the depth of the sediment which is irrigated, with larger values of $a$ resulting in more shallow bio-irrigation. Higher pumping rates, $r$, entail a faster removal of the tracer from the water. Compared to the parameters $r$ and $a$, the rate of adsorption, $k$ had a 1000-fold weaker effect on the outcome. Its value was set to 1 (h$^{-1}$) implying that it takes about 1 hour for the sediment adsorbed tracer fraction to be in equilibrium with the porewater tracer fraction.

In 11 out of 117 cases the fitting procedure yielded fits for which both the attenuation coefficient $a$ and the pumping rate $r$ were not significantly different from 0 and for which bio-irrigation was thus assumed to be absent. These were predominantly observed in November and December (7 out of 11 non-significant fits) and in these cases the tracer concentration did not notably change but rather fluctuated around a constant value. The fitted irrigation rates and attenuation coefficients did not show clear seasonal trends in the intertidal stations (Fig. 2). In the subtidal stations, irrigation rates were lowest in autumn, and

highest in winter (Fig. 2c). There was no significant difference in irrigation rates between the subtidal ($0.547 \pm 1.002$ mL cm$^{-2}$ h$^{-1}$) and intertidal ($0.850 \pm 1.157$ mL cm$^{-2}$ h$^{-1}$) (Welch two-sample T-test: $p = 0.708$). Seasonally averaged irrigation rates were highest at Lodijksegat in winter ($1.693 \pm 1.375$ mL cm$^{-2}$ h$^{-1}$), whereas in autumn at that same station they were lowest ($0.091 \pm 0.078$ mL cm$^{-2}$ h$^{-1}$). The model derived attenuation coefficients were significantly higher in the subtidal ($2.387 \pm 3.552$ cm$^{-1}$) than in the intertidal ($0.929 \pm 1.793$ cm$^{-1}$) (Welch two-sample T-test: $p = 0.041$).

From the irrigation rates, the mean individual irrigation rate, *irrdens*, was estimated by dividing the total pumping rate $r$ with the density of individuals. They varied between $3.167 \pm 2.878$ and $10.246 \pm 16.006$ mL ind$^{-1}$ h$^{-1}$ in the intertidal (summer and





spring resp.), and 8.272 ± 17.701 and 55.666 ± 139.942 mL ind$^{-1}$ h$^{-1}$ in the subtidal (autumn and spring resp., Table 4), a difference not found to be significant (Welch two-sample T-test: $p = 0.073$).

### 3.4 Co-inertia analysis

The first and second axes of the co-inertia analysis (CoiA) explained 57% and 19% of the variance in the dataset respectively (histogram inset Fig. 4a). Both the first and second axes of the MCA performed on the environmental parameters and of the PCA performed on the species community were correlated, indicated by high Pearson correlation coefficients (Figure a; for the first axis: $r = 0.95$, $p < 0.001$; for the second axis: $r = 0.92$, $p < 0.001$). This points to an overall high correspondence between both ordinations. In the MCA of the environmental variables, the first axis reflected mainly a grain size gradient from
very fine sandy to silty (Fig. 4b), with subtidal sites Lodijksegat (L) and Hammen (H) on the very fine sandy end, and the intertidal site Zandkreek (Z) in the high silt end (Fig. 4a). The Chl $a$ content and the immersion type (intertidal $vs$ subtidal) were the main factors associated with axis 2. This axis separated the subtidal station Viane (V) from the intertidal stations Dortsman (D) and Olzendenpolder (O). Of the different seasons, only summer correlated to the second axis. The PCA of the relative species abundances showed that in more fine sandy subtidal stations species such as the reef forming *Mytilus edulis*
(Linnaeus, 1758), and *Lanice conchilega* (Pallas 1766) were found (Fig. 4c). The species *Corophium sp.* and *Peringia ulvae* (Pennant, 1777) dominated in the intertidal, while *Ophiura ophiura* (Linnaeus, 1758) and *Nephtys hombergii* (Lamarck, 1818) were mainly found in the subtidal.

The correlation tests resulted in significant correlations between the first and the second axes of the co-inertia analysis (CoiA) for the BPc (axis 1: $r = 0.54$, $p = 0.008$; axis 2: $r = 0.65$, $p = < 0.001$), and the first CoiA axis and the IPc (axis 1: $r = 0.78$, $p =$
$< 0.001$; see Table 4 for full correlation statistics). Values for these indices are highest in the intertidal samples (Dortsman) and lowest in the subtidal, high Chl $a$ samples (Viane), where also respectively the highest and lowest species densities were recorded. For the individual irrigation rate *irrdens* and the attenuation coefficient $a$, only the correlation with the second axis was significant, with correlation values of -0.49 ($p = 0.017$) and -0.57 ($p = 0.005$) respectively. Both *irrdens* and the attenuation coefficient increased in the opposite direction of the BPc and IPc indices. No significant correlations were found for the model
derived pumping rate $r$ (axis 1: $r = -0.35$, $p = 0.107$; axis 2: $r = 0.263$, $p = < 0.226$). The pumping rate increased towards the intermediate – low Chl $a$ samples, almost perpendicular to both the IPc/BPc arrows and the attenuation coefficient (Fig. 5c).

## 4    Discussion

### 4.1  Bio-irrigation shape

Bio-irrigation is a complex process with profound effects on sediment biogeochemistry (Aller and Aller, 1998; Kristensen,
2001). For a better understanding of how bio-irrigation affects the sediment matrix, and to construct indices of irrigation based on species composition and life history traits, it is crucial to understand the mechanistic bases of the process. This is the first study in which continuous measurements of a tracer substance, and a mechanistic model have been combined to study the bio-





irrigation behaviour of species assemblages across a range of estuarine habitats. In such experiments, the tracer concentration in the overlying water decreases as it is diluted through mixing with porewater from the sediment. Initially, the sediment porewater is devoid of tracer, so that the dilution of the overlying water concentration is maximal. As the sediment itself becomes charged with tracer, the effect of sediment-water exchange on the bottom water concentration will decrease until the tracer concentration in the bio-irrigated part of the sediment and bottom water concentration are equal, and a quasi-steady state is achieved in which only molecular diffusion further slowly redistributes the tracer in the sediment. This verbal description of a bio-irrigation experiment shows that there are two important aspects to the data: the rate of bio-irrigation determines the initial decrease of tracer and how quickly the steady state will be reached, while the sediment depth over which bio-irrigation occurs determines the difference between initial and ultimate water column tracer concentrations at steady state.

The mechanistic model applied to our data comprises both these aspects, which are encompassed in two parameters: the integrated rate of bio-irrigation ($r$), and the attenuation coefficient ($a$) that determines the irrigation depth. In model simulations, the differences between fast and slow pumping rates mainly manifest themselves in the first part of the time series, while differences in irrigation depths are mainly discernable after several hours (Fig. 3b). This adds nuance to the interpretation of bio-irrigation rate, as similar irrigation rates may have divergent effects on sediment biogeochemistry when the depth over which solutes are exchanged differs. We have shown here that this nuance is at play in the Oosterschelde, where model derived pumping rates are very similar in subtidal and intertidal sediments, but the attenuation coefficient was higher for subtidal sites than for intertidal sites, implying a more shallow bio-irrigation pattern in the former.

Our tracer time series were measured at sufficiently high resolution (0.033 Hz), and for a sufficiently long time so that both the initial decrease, and the concentration to which the tracer converges were recorded. Indeed, identifiability analysis, a procedure to discover the certainty with which model parameters can be estimated from data (Soetaert and Petzoldt, 2010) showed that the information in our data was sufficient to estimate these two parameters ($r$ and $a$) with high confidence. This represents a significant improvement over discrete tracer measurements, from which deriving information of the depth distribution of irrigation is problematic (Andersson et al., 2006). Other data and/or models may not be able to derive these two quantities. Often bio-irrigation is estimated from linear fits through scarce ($\leq 5$ measurements) tracer concentration measurements (Mestdagh et al., 2018; De Smet et al., 2016; Wrede et al., 2018). This procedure is mainly applied when bromide is used as a tracer, as concentrations of this substance need to be measured in an elemental analyser, a procedure which for practical reasons does not allow for quasi-continuous measurements from the same sample. This has a major drawback, as the linearization of the exponential decrease will clearly underestimate the pumping rates, and it will be influenced by the (unknown) tracer depth (Fig. 3). Indeed, these methods are sensitive to the chosen duration of the experiment, e.g. as results based on a time series of 6 hours will not give the same results as those based on a 12 hour measurement.

### 4.2 Spatio-temporal variability in bio-irrigation

Our data show that although total pumping rates are similar in the subtidal and intertidal sediments of the Oosterschelde, irrigation is shallower in the subtidal, as indicated by the higher attenuation coefficient (Fig. 2: c, d). The species community



in the subtidal that is responsible for pumping is less dense, but (on average) biomass is higher than in the intertidal (Table 4). In Viane, the site where bioirrigation is lowest, only two species occur, *Ophiura ophiura* (Linneaus 1758), and *Nephtys hombergii*, and neither are typically associated with bioirrigation, although *O. ophiura* can significantly disturb the sediment surface, inducing shallow irrigation (Fig. 4c). The other two subtidal stations harbor two polychaetes species that have been

found to be prominent bio-irrigators: *Lanice conchilega* (Lodijksegat) and *Notomastus latericeus* (Sars 1851) (both Lodijksegat and Hammen). The sand mason worm *L. conchilega* lives in tubes constructed of shell fragments and sand particles which extend down to 10-15 cm (in the study area) and significantly affect the surrounding biogeochemistry (Forster and Graf, 1995; Braeckman et al., 2010). Highest densities of this species were observed in autumn at Lodijksegat, but interestingly this coincided with lowest bio-irrigation values for this station (Table 2: densities = $375 \pm 22$ ind m$^{-2}$; Fig. 2c: bio-irrigation = 0.091

$\pm$ 0.176 mL cm$^{-2}$ h$^{-1}$). High densities of *C. fornicata*, an epibenthic gastropod, in the same samples allow for the speculation that this species competes with infauna, suppressing the bio-irrigation behavior through constant agitation of the feeding apparatus, similar to what happens in non-lethal predator-prey interactions (Maire et al., 2010; De Smet et al., 2016). Burrows of *N. latericeus* extend down to 40 cm, and they have no lining, which –in theory- would facilitate irrigation. However, the burrows are considered semi-permanent, which in turn limits the depth up to which bio-irrigation plays a role (Kikuchi, 1987;

Holtmann et al., 1996). The presence of these polychaetes is thus not directly translated in high irrigation rates, though there does appear to be a logical link to the depth over which bio-irrigation occurs, with this being deepest in Lodijksegat (lowest *a*) where the species are present, and shallowest in Viane (highest *a*) that lacks these species.

In the intertidal stations the main species described as bio-irrigators are the mud shrimp *Corophium* sp*.*, the lugworm *Arenicola marina* (Linnaeus, 1758)*,* and the capitellid polychaete *Heteromastus filiformis* (Claparède, 1864). *Corophium sp.* is an active

bio-irrigator that lives in lined U-shaped burrows 5 to 10 cm in depth (McCurdy et al., 2000; De Backer et al., 2010). *A. marina* is often noted as the main bio-irrigator and bioturbator in marine intertidal areas (Huettel, 1990; Volkenborn et al., 2007). This species constructs U shaped burrows of 20 – 40 cm deep, and typically injects water to this depth in irrigation bouts of 15 minutes (Timmermann et al., 2007). *H. filiformis* creates mucus-lined permanent burrows in sediments up to 30 cm deep (Aller and Yingst, 1985). These species are present in all intertidal sites presented here. High densities of *Corophium sp.* are found

there where high irrigation rates are measured (table 2 and figure 2: Dortsman, occurring annually at $6781 \pm 5289$ ind m$^{-2}$ – bio-irrigation rates between $0.942 \pm 1.04$ and $1.149 \pm 0.645$ mL cm$^{-2}$ h$^{-1}$ along the year).

The higher abundance of previously mentioned bio-irrigators in the intertidal, as opposed to the subtidal, explains the lower attenuation coefficient values in the intertidal. When more individuals bio-irrigate over different depth ranges, this will impact the average irrigation depth more than a few individuals, even though the latter bio-irrigate deeper. Intertidal areas experience

stronger variations in physical stressors such as waves, temperature, light, salinity and precipitation than subtidal areas (Herman et al., 2001), and to biological stressors such as predation by birds (Fleischer, 1983; Granadeiro et al., 2006; Ponsero et al., 2016). Burrowing deeper, or simply residing in deeper sediment layers for a longer time, are valid strategies for species in the intertidal to combat these pressures (Koo et al., 2007; MacDonald et al., 2014).



### 4.3 The Bio-irrigation Potential

The Community Irrigation potential (Eq. 4, Wrede et al., 2018) subsumes both the depth of bio-irrigation and the rate. The former is represented by the injection depth (*ID*), while the latter relates to the burrowing (*BT*) and feeding type (*FT*) of the species traits scaled with their size and density. Interestingly, in the Oosterschelde data, only one of the irrigation parameters fitted in the model correlates to the IPc: the attenuation coefficient (Fig. 4c). This is most likely a logical consequence of the fact that the IPc index was calibrated using the $Br^-$ linear regression method (Wrede et al., 2018), which may mainly quantify

the irrigation depth. Nevertheless, the lack of a relation between the pumping rate and the IPc is surprising, and tends to suggest that bio-irrigation is a process which not only depends on the species characteristics but also includes context dependent trait modalities that need to be considered.

   Functional roles of species may differ depending on the context in which they are evaluated, and the a priori assignment of a species to a functional effect group may therefore be too simplistic (Hale et al., 2014; Murray et al., 2014). Christensen et al.

(2000) for instance reported irrigation rates of sediments in Kertinge Nor, Denmark with high abundances of *Hediste diversicolor* (O.F. Müller, 1776) (600 ind. $m^{-2}$ at 15 °C) that varied with a factor 4 whether the organism was suspension feeding (2704 $\pm$ 185 L $m^{-2}$ $d^{-1}$) or deposit-feeding (754 $\pm$ 80 L $m^{-2}$ $d^{-1}$). In our study, the intertidal station Zandkreek also had very high abundances of *H. diversicolor* (peak at 2550 ind. $m^{-2}$ in April) but much lower irrigation rates (128.6 $\pm$ 160.6 L $m^{-2}$ $d^{-1}$). Possibly, the higher Chl *a* concentrations in Zandkreek (20.2 µg $gDW^{-1}$) compared to the sediment in Christensen et al.

(2000) ($\pm7$ µg $gDW^{-1}$, converted from µg $gWW^{-1}$) caused the species to shift even more to deposit feeding. Similarly, previously reported irrigation rates of *Lanice conchilega* in late summer were quantified to range between 26.45 and 33.55 L $m^{-2}$ $d^{-1}$ (3243 $\pm$ 1094 ind $m^{-2}$, in an intertidal area in Boulogne-Sur-Mer, France (De Smet et al., 2016), whereas we measured rates that were more than an order of magnitude higher in the same season (229.3 $\pm$ 327.8 L $m^{-2}$ $d^{-1}$; Fig. 2c), although densities were an order of magnitude lower (298 $\pm$ 216 ind $m^{-2}$). *Lanice conchilega* is also known to switch from suspension-feeding to

deposit-feeding when densities are lower (Buhr, 1976; Buhr and Winter, 1977). This comparison suggests that bio-irrigation activity is higher when the *L. conchilega* is deposit feeding, although there could be of course additional context-dependent factors at play.

   The species community in which an organism occurs could also affect the bio-irrigation behavior. Species regularly compete for the same source of food (e.g. filter feeders), with species changing their feeding mode to escape competitive pressure

(Miron et al., 1992). If bio-irrigation is to provide oxygen or to reduce the build-up of metabolites, then, given sufficient densities of other bio-irrigating organisms, oxygen halo's may overlap (Dornhoffer et al., 2012), reducing the need for individuals to pump. In Zandkreek for instance, *Arenicola marina* (Linnaeus, 1758) was present in many samples, except during summer and autumn (Fig. 2b), while *Hediste diversicolor* was present in constant densities throughout the year. Although *A. marina* is a very vigorous bio-irrigator, its presence did not lead to a doubled pumping rate, suggesting an

adaptation of the pumping behaviour to the activity of *H. diversicolor*, or vice versa. This implies that simply summing of individual species irrigation scores to obtain a bio-irrigation rate may be too simplistic.



With these considerations in mind it appears that a comprehensive understanding of the ecology of species within the appropriate spatial scale and environmental context is a prerequisite for the application of an index to predict bio-irrigation rates (and by extension other functional traits). The current index (Eq. 4) contains burrow type, feeding mode, burrow depth, and an exponent to scale the metabolic rate, but from our analysis it appears that introducing more context-dependency could improve results. In Renz et al. (2018) for example, a distinction was made between an organism's activity based on the sediment type in which it occurred (cohesive or permeable sediment) in the calculation of their index, the Community Bioirrigation Potential (BIPc), although in this work no comparison with measured irrigation rates has taken place.

We stress the importance of measuring bio-irrigation rates in field settings, as it is through repeated measurements that the complex interactions of species communities and their environment will be best understood.

## 5    Conclusions

Benthic organisms differ strongly in the magnitude and mode in which they express functional traits. With this study we aimed to contribute to a better understanding of the mechanistic link between measured bio-irrigation rates and an ecological index. By fitting fluorescent tracer measurements using a mechanistic model we were able to infer more detailed information on the bio-irrigation process in species communities than an exchange rate alone, thereby improving on linear regression techniques. Similar irrigation rates may not have the same effects on the sediment biogeochemistry when the depth over which solutes are exchanged differs. Different assemblages of bio-irrigators may demonstrate this effect, as seen in this study. Our findings also highlight the importance of the context in which indices for functional processes should be evaluated, because of the confounding role spatial context and behaviour play.
**Code availability**

Model code and instructions for fitting similar data will be made available on request to the corresponding author.

**Author contribution**

E.D.B. developed the model and performed model simulations, performed statistical analysis, and prepared the manuscript
with contributions from all co-authors. J.T. collected field data, performed measurements, and analysed macrofauna. U.B. and
T.Y. contributed to the manuscript preparation. K.S. developed model code, and contributed to the manuscript.

**Competing interests**

The authors declare that they have no conflict of interest.

**Acknowledgements**

E.D.B. is a doctoral research fellow funded by the Belgian Science Policy Office (BELSPO) BELSPO, contract
BR/154/A1/FaCE-It. J.T. is a doctoral research fellow funded by the European Maritime and Fisheries Fund (EMFF), and the
Netherlands Ministry of Agriculture Nature and Food Quality (LNV) (Grant/Award Number: 1300021172). U.B. is a
postdoctoral research fellow at Research Foundation - Flanders (FWO, Belgium) (Grant 1201716N). We thank field
technicians, and laboratory staff: Pieter Van Rijswijk, Peter van Breugel and Yvonne van der Maas, as well as students that
assisted with the processing of samples: Paula Neijenhuys, Jolien Buyse, Vera Baerends. For the explanation of the ordination
methods we thank Olivier Beauchard. Lastly we thank the crew of the Research Vessel Delta.

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





**Figures**

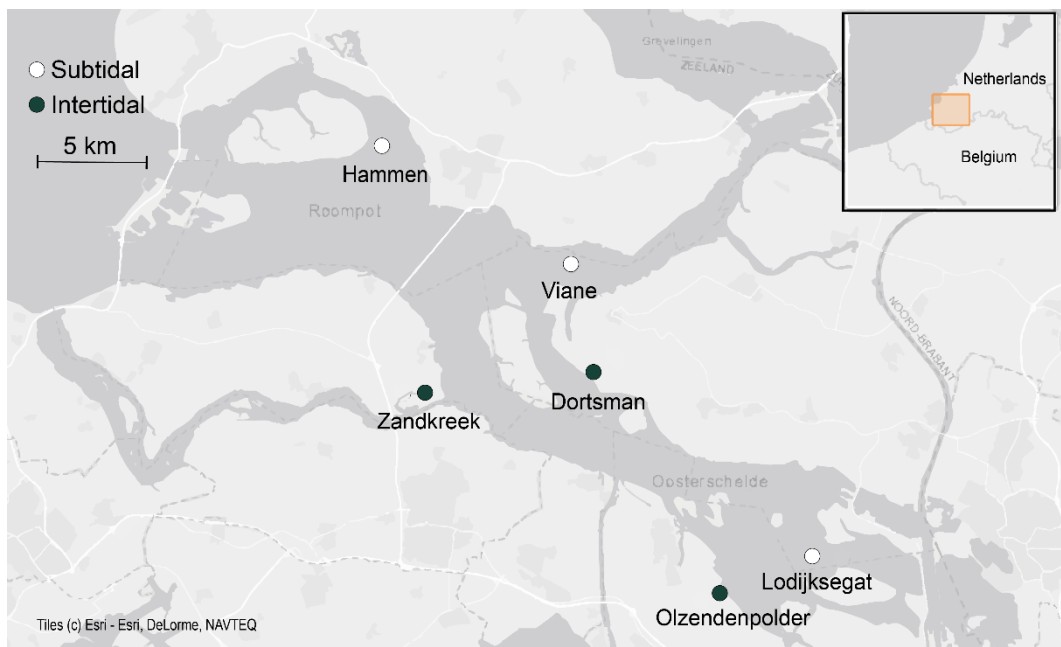

Figure 1: Overview of the subtidal (white dots) and intertidal (black dots) in the Oosterschelde estuary.



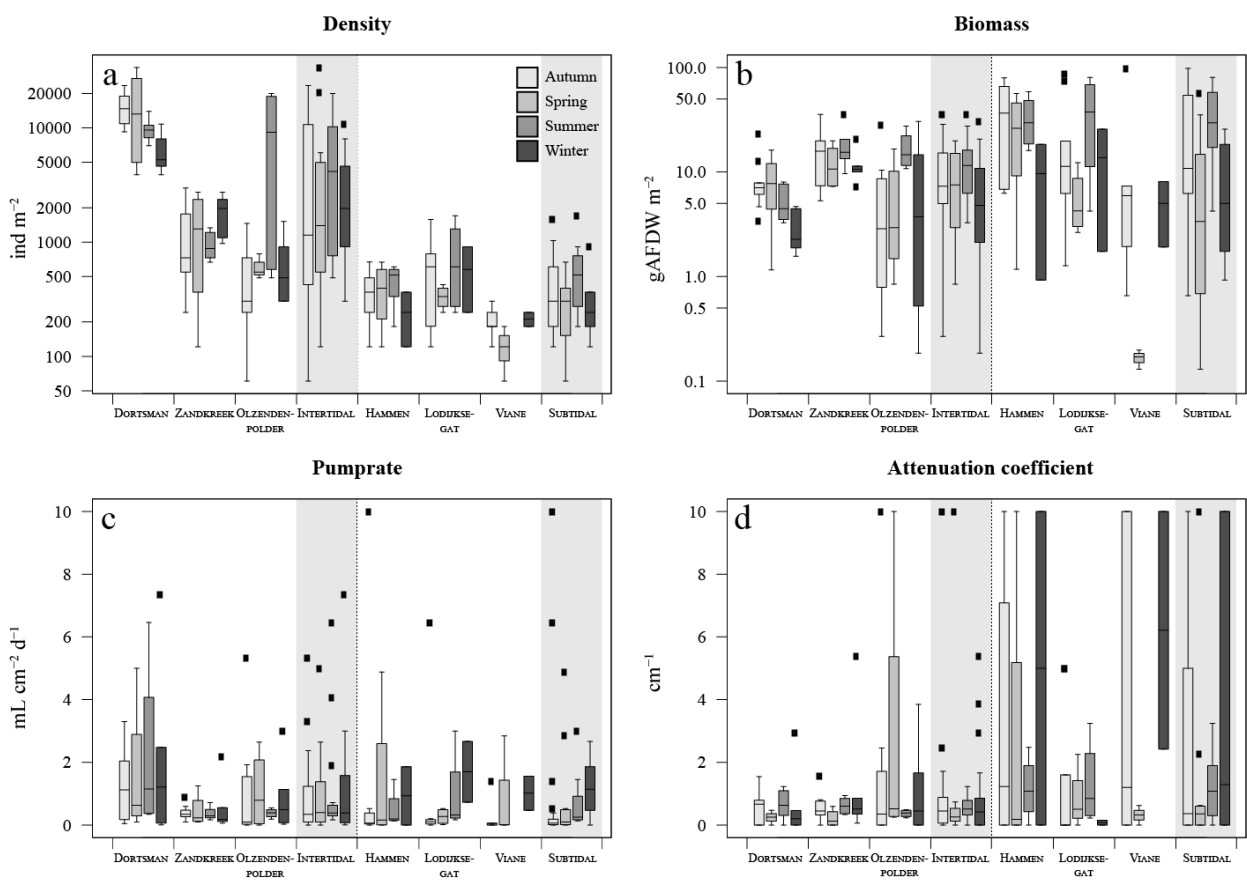


**Figure 2: Values for (a) organism densities (ind m-2); (b) organism ash-free dry weight (gAFDW m-2); (c) the model derived pumping rate (ml cm-2 d-1); (d) the model derived attenuation coefficient (cm-1) per station, and per habitat type (grey shaded areas, intertidal vs subtidal).**





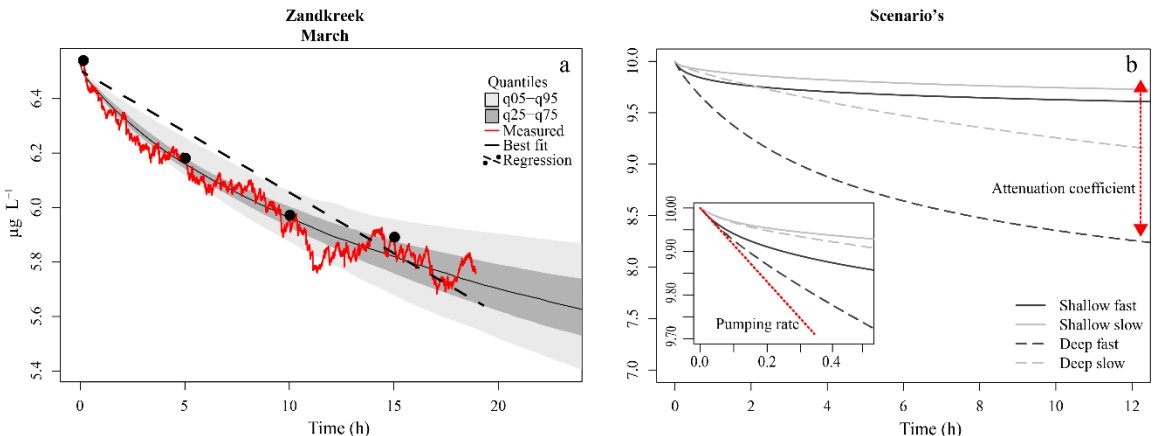

**Figure 3: (a) A model fit for data (red line) collected from a core at Zandkreek in March 2017. The best fit tracer profile (full black line) is shown, along with the range of model results as quantiles (light and dark grey). An example of a linear fit through samples taken every 5 hours is also shown. (b) Example model output for different combinations of pumping rate (slow = 0.15 ml cm$^{-2}$ h$^{-1}$ - fast = 0.8 ml cm$^{-2}$ h$^{-1}$), and attenuation coefficients (shallow = 5 - deep = 0.5 cm$^{-1}$). The inset shows a close-up of the first half hour of the simulation. Red lines illustrate the effects of the pumping rate, which has the strongest initial effect, and the attenuation coefficient, which determines the depth of the irrigation.**



**Figure 4: Summary of the coinertia analysis, and associated correlated response variables. (a) The co-structure between abiotic samples and species samples. The arrow origins are the environmental scores, the points are the species scores (light circles "D", "O","Z" for intertidal sites Dortsman, Olzendenpolder and Zandkreek respectively; dark circles "H", "L", "V" for subtidal sites Hammen, Lodijksegat and Viane respectively). (b) Multiple correspondence analysis for the environmental variables (MCA). (c) Principle component analysis (PCA) for the species, showing only those with ordination scores > 0.1. Gray arrows show the correlations between the response variables, and the axes of the coinertia analysis. The length of the arrows indicates the strength of the correlation.**






**Tables**

**Table 1: Sampling frequency of the different research sites.**

|  | Spring<br>Apr - Jun | Summer<br>Jul – Sep | Autumn<br>Oct - Dec | Winter<br>Jan - Mar |
|---|---|---|---|---|
| **Dortsman** | 4 | 5 | 9 | 5 |
| **Zandkreek** | 4 | 6 | 9 | 6 |
| **Olzendenpoder** | 4 | 4 | 8 | 6 |
| **Lodijksegat** | 4 | 4 | 8 | 2 |
| **Hammen** | 4 | 4 | 8 | 2 |
| **Viane** | 3 | 0 | 6 | 2 |

**Table 2: Sediment characteristics averaged over the study period (n= 8 per sampling site) represented with standard deviation for the intertidal sites Dortsman, Olzendenpolder and Zandkreek, and the subtidal sites Lodijksegat, Hammen and Viane.**

|  | Dortsman | Olzendenpolder | Zandkreek | Lodijksegat | Hammen | Viane |
|---|---|---|---|---|---|---|
| **% Silt** | $0 \pm 0$ | $14 \pm 16$ | $51 \pm 7$ | $25 \pm 5$ | $24 \pm 5$ | $63 \pm 19$ |
| **CN Ratio (-)** | $6.5 \pm 1.2$ | $11.3 \pm 2.4$ | $9.3 \pm 1.0$ | $12.4 \pm 1.4$ | $9.8 \pm 0.9$ | $9.9 \pm 1.0$ |
| **% $C_{org}$** | $0.07 \pm 0.02$ | $0.30 \pm 0.27$ | $0.79 \pm 0.33$ | $0.58 \pm 0.12$ | $0.35 \pm 0.07$ | $1.16 \pm 0.36$ |
| **MGS (µm)** | $140 \pm 2$ | $112 \pm 24$ | $59 \pm 14$ | $116 \pm 7$ | $201 \pm 38$ | $53 \pm 60$ |
| **Porosity (-)** | $0.43 \pm 0.07$ | $0.53 \pm 0.07$ | $0.45 \pm 0.09$ | $0.52 \pm 0.03$ | $0.45 \pm 0.03$ | $0.73 \pm 0.06$ |
| **_Chl_ a (µg g$^{-1}$)** | $8.65 \pm 3.53$ | $9.97 \pm 2.80$ | $20.60 \pm 4.19$ | $5.33 \pm 3.92$ | $3.76 \pm 2.43$ | $10.26 \pm 3.92$ |

**Table 3: Species occurrence per station and per season (ind m$^{-2}$), excluding species that were only encountered once.**

|  | Species | Autumn | Spring | Summer | Winter | Annual |
|---|---|---|---|---|---|---|
|  | _Arenicola marina_ | $113 \pm 74$ | $440 \pm 395$ | $91 \pm 35$ | 0 | $194 \pm 244$ |
|  | _Bathyporeia sp._ | $1789 \pm 1381$ | $3934 \pm 3087$ | $1443 \pm 1452$ | $577 \pm 350$ | $1735 \pm 1833$ |
|  | _Capitella capitata_ | $289 \pm 416$ | $223 \pm 153$ | $304 \pm 0$ | $73 \pm 27$ | $192 \pm 240$ |
|  | _Cerastoderma edule_ | $61 \pm 0$ | $61 \pm 0$ | $61 \pm 0$ | $81 \pm 35$ | $69 \pm 23$ |
|  | _Corophium sp._ | $9957 \pm 4465$ | $7120 \pm 9205$ | $5848 \pm 2792$ | $2977 \pm 1850$ | $6781 \pm 5289$ |
|  | _Eteone longa_ | $61 \pm 0$ | 0 | $122 \pm 0$ | $61 \pm 0$ | $85 \pm 33$ |
| **Dortsman** | _Hediste diversicolor_ | $91 \pm 61$ | $547 \pm 687$ | $304 \pm 182$ | $61 \pm 0$ | $243 \pm 311$ |
| **Intertidal** | _Limecola balthica_ | $122 \pm 0$ | 0 | $152 \pm 43$ | $61 \pm 0$ | $109 \pm 51$ |
|  | _Nematoda_ | 0 | $273 \pm 129$ | $61 \pm 0$ | 0 | $203 \pm 153$ |
|  | _Oligochaeta_ | $219 \pm 164$ | $851 \pm 0$ | $1175 \pm 1719$ | $122 \pm 50$ | $458 \pm 839$ |
|  | _Peringia ulvae_ | $1409 \pm 1538$ | $365 \pm 0$ | $658 \pm 729$ | $840 \pm 381$ | $911 \pm 933$ |
|  | _Pygospio elegans_ | $425 \pm 0$ | 0 | 0 | $61 \pm 0$ | $134 \pm 163$ |
|  | _Scoloplos armiger_ | $1782 \pm 1197$ | $1470 \pm 1195$ | $1288 \pm 691$ | $1580 \pm 970$ | $1572 \pm 1013$ |
|  | _Scrobicularia plana_ | $1175 \pm 460$ | $608 \pm 662$ | $759 \pm 301$ | $61 \pm 0$ | $753 \pm 570$ |



| | | | | | | |
|---|---|---|---|---|---|---|
| | *Tellinoidea* | $61 \pm 0$ | $61 \pm 0$ | $0$ | $61 \pm 0$ | $61 \pm 0$ |
| **Zandkreek** **Intertidal** | *Abra alba* | $76 \pm 30$ | $152 \pm 43$ | $91 \pm 43$ | $61 \pm 0$ | $95 \pm 44$ |
| | *Arenicola marina* | $61 \pm 0$ | $152 \pm 43$ | $0$ | $0$ | $122 \pm 61$ |
| | *Hediste diversicolor* | $1013 \pm 737$ | $1409 \pm 780$ | $1033 \pm 392$ | $1326 \pm 520$ | $1156 \pm 609$ |
| | *Heteromastus filiformis* | $0$ | $182 \pm 0$ | $0$ | $76 \pm 30$ | $97 \pm 54$ |
| | *Oligochaeta* | $324 \pm 175$ | $0$ | $0$ | $375 \pm 383$ | $358 \pm 316$ |
| | *Tharyx sp.* | $61 \pm 0$ | $0$ | $0$ | $91 \pm 43$ | $81 \pm 35$ |
| **Olzendenpolder** **Intertidal** | *Arenicola marina* | $142 \pm 93$ | $122 \pm 105$ | $122 \pm 105$ | $122 \pm 0$ | $128 \pm 83$ |
| | *Capitella capitata* | $61 \pm 0$ | $101 \pm 35$ | $61 \pm 0$ | $0$ | $85 \pm 33$ |
| | *Cerastoderma edule* | $61 \pm 0$ | $61 \pm 0$ | $61 \pm 0$ | $0$ | $61 \pm 0$ |
| | *Crangon crangon* | $0$ | $61 \pm 0$ | $122 \pm 0$ | $0$ | $76 \pm 30$ |
| | *Hediste diversicolor* | $61 \pm 0$ | $61 \pm 0$ | $0$ | $182 \pm 0$ | $122 \pm 70$ |
| | *Heteromastus filiformis* | $0$ | $122 \pm 0$ | $0$ | $61 \pm 0$ | $101 \pm 35$ |
| | *Notomastus sp.* | $81 \pm 35$ | $61 \pm 0$ | $61 \pm 0$ | $152 \pm 78$ | $108 \pm 66$ |
| | *Oligochaeta* | $0$ | $122 \pm 0$ | $152 \pm 43$ | $213 \pm 215$ | $170 \pm 117$ |
| | *Peringia ulvae* | $61 \pm 0$ | $0$ | $12454 \pm 10795$ | $304 \pm 86$ | $6339 \pm 9566$ |
| | *Polydora ciliata* | $122 \pm 0$ | $0$ | $0$ | $61 \pm 0$ | $101 \pm 35$ |
| | *Scoloplos armiger* | $344 \pm 220$ | $410 \pm 135$ | $182 \pm 105$ | $279 \pm 213$ | $314 \pm 188$ |
| | *Tharyx sp.* | $243 \pm 61$ | $0$ | $0$ | $61 \pm 0$ | $152 \pm 107$ |
| **Hammen** **Subtidal** | *Actiniaria* | $144 \pm 72$ | $97 \pm 54$ | $134 \pm 51$ | $61 \pm 0$ | $125 \pm 62$ |
| | *Ensis sp.* | $61 \pm 0$ | $0$ | $61 \pm 0$ | $0$ | $61 \pm 0$ |
| | *Hemigrapsus sp.* | $61 \pm 0$ | $0$ | $122 \pm 0$ | $0$ | $81 \pm 35$ |
| | *Mytilus edulis* | $61 \pm 0$ | $3311 \pm 215$ | $2886 \pm 2105$ | $0 \pm 0$ | $2491 \pm 1735$ |
| | *Nephtys hombergii* | $85 \pm 33$ | $61 \pm 0$ | $61 \pm 0$ | $61 \pm 0$ | $71 \pm 24$ |
| | *Notomastus sp.* | $111 \pm 81$ | $203 \pm 93$ | $152 \pm 43$ | $61 \pm 0$ | $137 \pm 82$ |
| | *Ophiura ophiura* | $122 \pm 0$ | $0$ | $243 \pm 161$ | $0$ | $213 \pm 145$ |
| | *Scoloplos armiger* | $0$ | $61 \pm 0$ | $0$ | $91 \pm 43$ | $81 \pm 35$ |
| | *Terebellidae* | $61 \pm 0$ | $61 \pm 0$ | $61 \pm 0$ | $0 \pm 0$ | $61 \pm 0$ |
| **Lodijksegat** **Subtidal** | *Crepidula fornicata* | $319 \pm 152$ | $122 \pm 0$ | $972 \pm 172$ | $0$ | $477 \pm 369$ |
| | *Hemigrapsus sp.* | $61 \pm 0$ | $0$ | $61 \pm 0$ | $0$ | $61 \pm 0$ |
| | *Lanice conchilega* | $375 \pm 225$ | $304 \pm 0$ | $91 \pm 43$ | $273 \pm 301$ | $298 \pm 216$ |
| | *Malmgrenia darbouxi* | $91 \pm 43$ | $0$ | $0$ | $182 \pm 0$ | $122 \pm 61$ |
| | *Nephtys hombergii* | $111 \pm 60$ | $158 \pm 92$ | $0$ | $0$ | $133 \pm 76$ |
| | *Notomastus sp.* | $81 \pm 35$ | $91 \pm 43$ | $61 \pm 0$ | $61 \pm 0$ | $78 \pm 30$ |
| | *Pholoe baltica* | $61 \pm 0$ | $0$ | $122 \pm 0$ | $61 \pm 0$ | $76 \pm 30$ |
| | *Scoloplos armiger* | $122 \pm 0$ | $61 \pm 0$ | $122 \pm 0$ | $122 \pm 0$ | $106 \pm 30$ |
| | *Terebellidae* | $31 \pm 42$ | $0$ | $0$ | $61 \pm 0$ | $41 \pm 34$ |
| **Viane** **Subtidal** | *Nephtys hombergii* | $162 \pm 93$ | $101 \pm 70$ | $0$ | $122 \pm 0$ | $129 \pm 68$ |
| | *Ophiura ophiura* | $167 \pm 58$ | $0$ | $0$ | $91 \pm 43$ | $142 \pm 63$ |



**Table 4: Seasonally averaged values for Chl a in the upper 2 cm of the sediment (µg Chl a g-1), species density (ind m-2), biomass (gAFDW m-2), pumping rate (ml cm-2 h-1), attenuation coefficient (cm-1) and the individual irrigation rate (irrdens, ml ind-1 h-1) for the intertidal and the subtidal.**

|  | Season | Chl *a* | Density | Biomass | Pump rate | Attenuation | Irrdens |
|---|---|---|---|---|---|---|---|
| **Intertidal** | Autumn | 12.49 ± 6.92 | 5828 ± 7509 | 11.16 ± 9.31 | 0.88 ± 1.24 | 0.97 ± 1.91 | 9.32 ± 23.98 |
|  | Spring | 12.30 ± 3.89 | 6005 ± 10421 | 8.72 ± 6.48 | 1.03 ± 1.48 | 1.09 ± 2.81 | 10.25 ± 16.01 |
|  | Summer | 14.69 ± 6.58 | 6193 ± 6763 | 13.65 ± 8.91 | 0.72 ± 1.02 | 0.59 ± 0.34 | 3.17 ± 2.88 |
|  | Winter | 14.17 ± 7.52 | 2645 ± 2702 | 8.02 ± 8.10 | 0.79 ± 0.96 | 1.05 ± 1.56 | 9.07 ± 16.03 |
| **Subtidal** | Autumn | 5.90 ± 4.37 | 439 ± 365 | 25.67 ± 30.42 | 0.16 ± 0.31 | 2.96 ± 3.91 | 8.27 ± 17.70 |
|  | Spring | 7.00 ± 3.00 | 298 ± 181 | 12.15 ± 18.08 | 0.83 ± 1.58 | 1.33 ± 2.95 | 55.67 ± 139.94 |
|  | Summer | 4.20 ± 2.27 | 623 ± 494 | 36.67 ± 26.29 | 0.73 ± 1.02 | 1.23 ± 1.14 | 11.04 ± 9.54 |
|  | Winter | 6.02 ± 7.08 | 344 ± 289 | 9.45 ± 10.32 | 1.22 ± 0.99 | 3.76 ± 4.92 | 43.16 ± 40.80 |


**Table 5: Pearson correlations of the response variables against the ordination axes of the coinertia analysis, with p values reported under the values in italics.**

|  | Irrdens | Irrigation *r* | Attenuation *a* | BPc | IPc |
|---|---|---|---|---|---|
|  | ml ind$^{-1}$ h$^{-1}$ | ml cm$^{-2}$ h$^{-1}$ | cm$^{-1}$ | gWW$^{0.5}$ m$^{-2}$ | gAFDW$^{0.75}$ m$^{-2}$ |
| **Axis 1** | -0.194 | -0.345 | -0.288 | 0.540 | 0.780 |
|  | *0.375* | *0.107* | *0.182* | *0.008* | *< 0.001* |
| **Axis 2** | -0.493 | 0.263 | -0.565 | 0.646 | 0.395 |
|  | *0.017* | *0.226* | *0.005* | *< 0.001* | *0.062* |