# Peer review of "Biological and biogeochemical methods for estimating bio-irrigation: a case study in the Oosterschelde estuary."

_Biogeosciences, 2019_

## Referee Comment (RC1) · Anonymous Referee #1 · 22 Nov 2019

In this study, De Borger et al. investigate the irrigation activity of macrofaunal communities inhabiting estuarine mudflats of the Oosterschelde. Beyond the seasonal quantification of bioirrigation processes and the comparison of subtidal and intertidal ecosystems, the authors evaluate/validate the capacity of the community irrigation potential (IPc) developed by Wrede et al. 2018 to estimate both bioirrigation depth and rate based on biological traits. In light of their results, they conclude that the IPc give acceptable predictions of the bioirrigation depth but not bioirrigation rate. I found this paper a credible and interesting contribution with, nevertheless, a real uncertainty concerning the methodology used to quantify bioirrigation rates. Overall, the text is well written and informative and the conclusion drawn from the study is consistent with the

intention of the paper.

General comments: - My major concern deals with the use of uranine as dissolved tracer to quantify bioirrigation rates. Indeed, it is well known that uranine easily adsorbs to organic material so that bioirrigation rates can be severely overestimated if this process is not accurately quantified. To this end, the authors performed batch experiments to estimate the adsorption of uranine to the mud (i.e. sedimentary organic matter) but, if I understand well, they did not take into account the capacity of suspension-feeding bivalves to decrease uranine concentration in the overlying water through their filtration activity. I know from my own experience that bivalves such as Cerastoderma edule, Scrobicularia plana, Mytilus edulis or the filter-feeding gasteropod Crepidula fornicata are able to rapidly trap a large amount of uranine in their mantle cavity (on the gill surfaces). Given that these species were very abundant in some sampling stations (see Table 3), what is the level of accuracy of bioirrigation rates measured in the corresponding experimental cores?

- The comparison between experimentally measured bioirrigation rates and traits-based ecological indices should be discussed a little bit further.

Specific comments: - Abstract: Line 11-12: Biological traits do not really allow for the quantification of bioirrigation. This trait-based index (IPc) only give a more or less "rough" prediction of bioirrigation depth and rate. The sentence line 42 seems more correct. Line 16-17: I well understand that irrigation rates can be significantly affected by bioirrigator densities but it is not clear to me how higher densities could impact the bioirrigation depth.

- Introduction: Line 27: How do bioirrigation increase the exchange surface? Line 40: The term "pumping activity" is usually employed to describe the "filtration activity" in suspension-feeding bivalves. May "ventilation activity" be more appropriate? Line 42: Yes but over very different spatial scales.

- Materials and methods Line 74: I am wondering whether the sampling method (i.e.

small sediment cores <20 cm) really allows for the collection all bioirrigators inhabiting intertidal mudflats of the Oosterschelde estuary. It is clear that large and/or deep infaunal species such as burrowing mud shrimp could not be properly captured, thus leading to strong underestimation of bioirrigation rates. Line 80: How long have experimental cores been kept in buffering seawater tanks before the beginning of the experiments? Line 81: What was the average temperature at each studied season? That's a very important factor, which greatly determines the activity level of benthic invertebrates. Line 128-129: Batch adsorption experiments have been performed with sediment cores collected from Zandkreek (%Corg=0.79) and Dortsman (%Corg=0.07). Why not with sediment cores from Viane where the proportion of organic carbon is the highest (potentially the highest adsorption rate).

- Results Bioirrigation rates: The "pumping" rate and the irrigation attenuation coefficient were estimated by fitting a mathematical model to the tracer data. However, I'm wondering if the first part of the experiments (10-20 min) should be considered for the estimation of these parameters as the initial decrease in uranine concentration may mainly result from its rapid adsorption onto surficial organic particles as well as the mixing between overlying and burrow waters (which is not really sediment bioirrigation from my point of view).

- Discussion Line 281: I don't really understand the conclusion stating that the density of bioirrigating species would affect more the bioirrigation depth than the bioirrigation rates. I would have believed the opposite. Indeed, it has been reported that faunal activities (e.g. feeding, burrowing) can be altered by intense intraspecific interactions leading to lower bioturbation rates. Is there any references showing that increasing invertebrate densities result in increasing bioirrigation depths? Line 290-292: Another hypothesis is that high densities of C. fornicata may induce a rapid deposition of fine particles at the sediment-water interface (i.e. biodeposition) thus decreasing the permeability of upper sediment layers. Line 293: Burrows of N. latericeus can extend down to 40 cm, yet experimental chambers were only 20 cm long. Thus, the ventilation

activity of the worms may have been biased due to a constrained (shallow) benthic habitat. Lines 307-309: See comment Line 281. Why organisms of the same species (same stage) would irrigate over different depth ranges at high densities?

---

## Author Comment (AC1) · 2 Dec 2019

We thank the reviewer for taking the time to read the manuscript and provide thoughtful and constructive feedback. This feedback has highlighted areas which needed improvement in the form of additional references, or further clarification, and we were happy to improve these in the manuscript. The changes were applied in track changes in the manuscript, and copied in this reply in our answers. We also hope to have clarified the additional remarks about the methodology in a satisfactory manner. The reply itself was uploaded as a supplement.

[Figure]

Please also note the supplement to this comment:
https://www.biogeosciences-discuss.net/bg-2019-413/bg-2019-413-AC1-
supplement.pdf

**Supplement:**

We thank the reviewer for taking the time to read the manuscript and provide thoughtful and constructive feedback. This feedback has highlighted areas which needed improvement in the form of additional references, or further clarification, and we were happy to improve these in the manuscript. The changes were applied in track changes in the manuscript, and copied in this reply in our answers. We also hope to have clarified the additional remarks about the methodology in a satisfactory manner.

**1. Reviewer's comment:** General comments: - My major concern deals with the use of uranine as dissolved tracer to quantify bioirrigation rates. Indeed, it is well known that uranine easily adsorbs to organic material so that bioirrigation rates can be severely overestimated if this process is not accurately quantified. To this end, the authors performed batch experiments to estimate the adsorption of uranine to the mud (i.e. sedimentary organic matter) but, if I understand well, they did not take into account the capacity of suspension-feeding bivalves to decrease uranine concentration in the overlying water through their filtration activity. I know from my own experience that bivalves such as Cerastoderma edule, Scrobicularia plana, Mytilus edulis or the filter-feeding gasteropod Crepidula fornicata are able to rapidly trap a large amount of uranine in their mantle cavity (on the gill surfaces). Given that these species were very abundant in some sampling stations (see Table 3), what is the level of accuracy of bioirrigation rates measured in the corresponding experimental cores?

**Reply:** We had a similar concern during the initial testing of this method, therefore we performed an experiment. Six live *C. edule* were placed into incubation cores (performed in duplo, same controlled setup as in the manuscript) without sediment, and the tracer concentrations were monitored over time. Figure R1(a) shows the results of that trial, and from this we concluded that there was no obvious interaction between the organisms, and the uranine concentration. We also tested whether the cores, stirring devices, or bubbling stones interacted with the uranine, by monitoring the tracer profile in cores filled with only water with- and without a bubbling stone (Figure R1(b)). No reports of uranine adsorption/trapping by organisms were found in literature to warrant further testing. However, we would be interested in seeing results of your experiments for future reference. We added this figure in the supplement, and referred to it on line 87 of materials and methods: "Short experiments were performed to assess possible interactions between the tracer, and the used setup (Supplement)".

[Figure]

**Figure R1: a. Concentration of uranine over time in duplicate cores containing six live cockles each. b. Two cores aerated with a bubbling stone (black), and one non-aerated core (gray). For the aerated core two concentrations were tested to assess whether this would have any impact.**

**2. Reviewer's comment:** The comparison between experimentally measured bioirrigation rates and traitsbased ecological indices should be discussed a little bit further.

**Reply:** On your suggestion we expanded the discussion on lines 348-351, with the inclusion of predator – prey interactions: "Species also compete in the form of predator prey interactions, which have also been shown to alter behavior. For example, the presence of *Crangon crangon* reduced the food uptake of *L. conchilega* (De Smet et al., 2016), and altered the sediment reworking mode of *L. balthica* (Maire et al., 2010), in both cases because *C. crangon* predates on the feeding apparatus of these species protruding from the sediment."; and on lines 365-371 with macrofaunal responses to seasonal temperature variations: "Furthermore, Wrede et al. (2018) suggested to include

a temperature correction factor ($Q_{10}$) in the calculations to account for the expected metabolic response of macrofauna to increasing water temperatures (Brey, 2010). This temperature effect on benthic activity has indeed been noticed in similar works (Magni and Montani, 2006; Rao et al., 2014), but in this study and others the highest temperatures were not clearly associated with highest functional process rates (Schlüter et al., 2000: Braeckman et al., 2010; Queirios et al., 2015). The reasons for this ranged from a mismatch between food availability and the temperature peak , upward migration in the sediment to escape hypoxia, or the presence of confounding factors in the analysis such as faunal abundances and behavior (Forster et al., 2003)".

**3. Reviewer's comment:** Specific comments: - Abstract: Line 11-12: Biological traits do not really allow for the quantification of bioirrigation. This trait-based index (IPc) only give a more or less "rough" prediction of bioirrigation depth and rate. The sentence line 42 seems more correct.

**Reply:** Agreed, we added the word "estimate" to indicate this, line 11-12 now reads: "Quantification of bio-irrigation is done either through measurements with tracers, or more recently, estimated using biological traits to derive the community (bio-) irrigation potential (IPc)."

**4. Reviewer's comment:** Line 16-17: I well understand that irrigation rates can be significantly affected by bioirrigator densities but it is not clear to me how higher densities could impact the bioirrigation depth.

**Reply:** Please see our reply to your similar comment no. 13.

**5. Reviewer's comment:** Introduction: Line 27: How do bioirrigation increase the exchange surface?

**Reply:** When there are no burrowing organisms, the sediment-water interface (SWI) can be considered flat with a certain surface area per m² of seafloor. When organisms construct burrows connected to the SWI, the surface area is extended in the vertical dimension, as the burrow walls are in connection with the overlying water. In the cited study (Quintana et al., 2007), the effect of two burrowing polychaetes on solute transport was investigated. One species (*M. viridis*) caused higher exchange through increasing diffusion rates, the other (*H. filiformis*) caused higher exchange rates through nonlocal exchange. The flux of oxygen is increased as the polychaetes consume it, but the burrow walls are an additional (anoxic) surface over which oxygen can be taken up, hence an extension of the exchange surface. To clarify this matter, we partially integrated this explanation in the introduction on lines 26 – 29: "By extending the otherwise horizontal sediment water interface in the vertical dimension, burrowing organisms extend the exchange surface, especially when the burrow water is refreshed by ventilation activities. This enhances nutrient exchange (Quintana et al., 2007), and increases degradation rates (Na et al., 2008)."

**6. Reviewer's comment:** Line 40: The term "pumping activity" is usually employed to describe the "filtration activity" suspension-feeding bivalves. May "ventilation activity" be more appropriate?

**Reply:** In reading a significant part of the literature concerning bio-irrigation, we have not encountered this distinction, but it's an interesting remark. There seems to be no real consistency in terminology, and the preference of "pumping" vs. "ventilation" differs between authors (just a few examples: pumping: Berg et al. (2003), Forster and Graf, (1995); ventilation: Stief and de Beer (2006), Kristensen et al. (2012), or both: Renz et al. (2018), Roskosch et al. (2011)). To us the most "correct", based on consensus, is that bio-irrigation is the effect (enhanced pore-water transport and exchange etc.), and that it is caused by ventilation activity in general, with pumping activity as the main mechanism (Kristensen et al., 2012; Meysman et al., 2006). In this manuscript the terminology ("pumping rate") used in the rest of the text, reflects the initial modelling of *A. marina* as a "pump" by Meysman et al. (2006), we would like to keep it as such.

**7. Reviewer's comment:** Line 42: Yes but over very different spatial scales.

**Reply:** If we interpreted your comment correctly (that indices are most valid to compare bio-irrigation over larger spatial scales), we agree with you. However, the methodology in Wrede et al. (2018) rightly compares index values to measured irrigation rates from the same sediment cores, thus on the same scale.

**8. Reviewer's comment:** Materials and methods Line 74: I am wondering whether the sampling method (i.e. small sediment cores <20 cm) really allows for the collection all bioirrigators inhabiting intertidal mudflats of the Oosterschelde estuary. It is clear that large and/or deep infaunal species such as burrowing mud shrimp could not be properly captured, thus leading to strong underestimation of bioirrigation rates.

**Reply:** This is a valid remark on the limitations of this study as larger individuals of *A. marina* and *N. latericeus* were possibly omitted from the sampling. The chosen cores were a trade-off between the sediment surface that could be sampled, and how deep it could be sampled, as cores that are both deep (50 cm), and wide enough (20 cm) would compromise the feasibility of successfully collecting samples in the field. We have added an acknowledgment of this fact in the discussion on lines 266-269: "It should be noted that given the maximal depth of the incubation chambers was 20 cm of sediment, individuals of some species living deeper that this were not included in the incubations, and thus measurements of bio-irrigation (e.g. larger *A. marina*, or *N. latericeus*). This means that the patterns described previously can only be applied with certainty to the upper 20 cm of the sediment."

**9. Reviewer's comment:** Line 80: How long have experimental cores been kept in buffering seawater tanks before the beginning of the experiments?

**Reply:** As stated in materials and methods (line 83), the experimental cores were kept for 24 – 48 hours, in buffering water tanks before the start of the experiment.

**10. Reviewer's comment:** Line 81: What was the average temperature at each studied season? That's a very important factor, which greatly determines the activity level of benthic invertebrates.

**Reply:** The reviewer is correct. Average temperatures of the water in the cores during measurements were added to table 1, and referred to in methods on line 81.

| Season | Spring | Summer | Autumn | Winter |
|---|---|---|---|---|
| Months | Apr – Jun | Jul – Sep | Oct – Dec | Jan – Mar |
| Avg. Temperature (°C) | 12.8 | 17.9 | 11.9 | 7.3 |
| Dortsman | 4 | 5 | 9 | 5 |
| Zandkreek | 4 | 6 | 9 | 6 |
| Olzendenpoder | 4 | 4 | 8 | 6 |
| Lodijksegat | 4 | 4 | 8 | 2 |
| Hammen | 4 | 4 | 8 | 2 |
| Viane | 3 | 0 | 6 | 2 |

**11. Reviewer's comment:** Line 128-129: Batch adsorption experiments have been performed with sediment cores collected from Zandkreek (%Corg=0.79) and Dortsman (%Corg=0.07). Why not with sediment cores from Viane where the proportion of organic carbon is the highest (potentially the highest adsorption rate).

**Reply:** The final methodology for determining the adsorption coefficient was established only very late during the sampling period. Previous trials were all based on batch adsorption experiments with dried sediment (Gerke et al., 2008). However, we figured out that these readings were very much affected by the pH dependence of the fluorescence of uranine (see Gerke et al. (2013), and Figure R2, our own measurements of uranine fluorescence under a pH range). The degradation of organic matter present in the dried sediment, in the enclosed small volumes of the batch adsorption experiments, lowered the pH of the water outside of the stable range for uranine. This prompted the shift to working with natural sediment cores, where the pH is stable within the range "safe to use"

for uranine. However, by this time we had no more sediment cores from the Viane station, and retrieving them was impossible as these subtidal samples were taken onboard of a vessel outside of the research institute. Given the large difference between organic carbon values acquired for Dortsman (%Corg = 0.07 ±) and Zandkreek (%Corg=0.79 ± 0.33), and the similarity between Zandkreek and Viane in terms of MGS (59 ± 14 – 53 ± 60 μm), silt content (51 ± 7 – 63 ± 19 %), and OM content (0.79 ± 0.33 - 1.16 ± 0.36), we believe that actual values measured in Viane would not significantly differ from those of Zandkreek. Incidentally chl *a* values (also a measure for organic matter content) were highest in Zandkreek (20.60 ± 4.19 μ g$^{-1}$).

[Figure]

**Figure R2: measurements of uranine concentration measured (points) vs. fluorescence predicted based on added uranine concentration. Note that the lines are not straight, as the added acidic solution (0.62 M HCl) was added dropwise, which eventually decreases the concentration of the tracer.**

**12. Reviewer's comment:** Results Bioirrigation rates: The "pumping" rate and the irrigation attenuation coefficient were estimated by fitting a mathematical model to the tracer data. However, I'm wondering if the first part of the experiments (10-20 min) should be considered for the estimation of these parameters as the initial decrease in uranine concentration may mainly result from its rapid adsorption onto surficial organic particles as well as the mixing between overlying and burrow waters (which is not really sediment bioirrigation from my point of view).

**Reply:** The way the stirring device was set up, mixing of the uranine in the water column took place in about two minutes of injection of the tracer (Fig. R3). This period was always cut out of the analysis. However, we argue that from then onwards we see the effects of burrow ventilation. Water in burrows is exchanged through organism activities in this timeframe, as diffusive exchange would take much longer. Additionally, often cores showed a flat profile from the start onwards, regardless of the organic matter or mud content.

[Figure]

**Figure R3: First minutes of an incubation, after adding the uranine spike. Mixing takes place within 2-3 minutes, after which the starting concentrations is considered stable, and when the incubation starts.**

**13. Reviewer's comment:** Discussion Line 281: I don't really understand the conclusion stating that the density of bioirrigating species would affect more the bioirrigation depth than the bioirrigation rates. I would have believed the opposite. Indeed, it has been reported that faunal activities (e.g. feeding, burrowing) can be altered by intense intraspecific interactions leading to lower bioturbation rates. Is there any references showing that increasing invertebrate densities result in increasing bioirrigation depths?

**Reply:** We could not find studies where the depth of bio-irrigation is directly measured in combination with varying densities and species compositions. However, in Braeckman et al. (2010), increasing densities of *L. conchilega* tended to increase oxygen penetration, likely due to bio-irrigation. Our conclusion is twofold: in the intertidal, species that are assumed to be "deep" irrigators (> 5-10 cm, in terms of our model) are present in higher densities, and despite their smaller individual sizes (intertidal: higher densities, for similar biomass), we expect the effect of deeper burrows to still be noticeable in the tracer profiles. In addition to this, species in the intertidal have been found to reside deeper in the sediment, to escape the different pressures associated with this habitat (references in manuscript on lines 318-320).

**14. Reviewer's comment:** Line 290-292: Another hypothesis is that high densities of C. fornicata may induce a rapid deposition of fine particles at the sediment-water interface (i.e. biodeposition) thus decreasing the permeability of upper sediment layers.

**Reply:** This is indeed an interesting hypothesis, we have added it to the text with the appropriate references (line 298 - 300): "*C. fornicata* is also known to cause significant biodeposition of fine particles on the sediment surface (Ehrhold et al., 1998; Ragueneau et al., 2005). This could decrease the permeability of the surface layers and as such decrease the extent of possible bio-irrigation."

**15. Reviewer's comment:** Line 293: Burrows of N. latericeus can extend down to 40 cm, yet experimental chambers were only 20 cm long. Thus, the ventilation activity of the worms may have been biased due to a constrained (shallow) benthic habitat.

**Reply:** We acknowledge this, and as discussed in a previous comment, a paragraph has been added on limitations of the study.
Lines 266 – 269: It should be noted that given the maximal depth of the incubation chambers was 20 cm of sediment, individuals of some species living deeper that this were not included in the incubations, and thus measurements of bio-irrigation (e.g. larger *A. marina*, or *N. latericeus*). This means that the patterns described previously can only be applied with certainty to the upper 20 cm of the sediment.

**16. Reviewer's comment:** Lines 307-309: See comment Line 281. Why organisms of the same species (same stage) would irrigate over different depth ranges at high densities?

**Reply: Please see our reply to comment 13**

**References used in this reply.**
Berg, P., Røy, H., Janssen, F., Meyer, V., Jørgensen, B. B., Huettel, M. and De Beer, D.: Oxygen uptake by aquatic sediments measured with a novel non-invasive eddy-correlation technique, Mar. Ecol. Prog. Ser., 261, 75–83, doi:10.3354/meps261075, 2003.
Braeckman, U., Provoost, P., Gribsholt, B., Van Gansbeke, D., Middelburg, J. J., Soetaert, K., Vincx, M. and Vanaverbeke, J.: Role of macrofauna functional traits and density in biogeochemical fluxes and bioturbation, Mar. Ecol. Prog. Ser., 399(2010), 173–186, doi:10.3354/meps08336, 2010.
Brey, T.: An empirical model for estimating aquatic invertebrate respiration, Methods Ecol. Evol., 1(1), 92–101, doi:10.1111/j.2041-210x.2009.00008.x, 2010.
Ehrhold, A., Blanchard, M., Auffret, J.-P. and Garlan, T.: Conséquences de la prolifération de la crépidule (Crepidula fornicata) sur l'évolution sédimentaire de la baie du Mont-Saint-Michel (Manche, France), Comptes

Rendus l'Académie des Sci. - Ser. IIA - Earth Planet. Sci., 327(9), 583–588, doi:https://doi.org/10.1016/S1251-8050(99)80111-6, 1998.

Forster, S. and Graf, G.: Impact of irrigation on oxygen flux into the sediment: intermittent pumping by Callianassa subterranea and "piston-pumping" by Lanice conchilega, Mar. Biol., 123(2), 335–346, doi:10.1007/BF00353625, 1995.

Forster, S., Khalili, A. and Kitlar, J.: Variation of nonlocal irrigation in a subtidal benthic community, , (1980), 335–357, 2003.

Gerke, K. M., Sidle, R. C. and Tokuda, Y.: Sorption of Uranine on Forest Soils, Hydrol. Res. Lett., 2(May), 32–35, doi:10.3178/hrl.2.32, 2008.

Gerke, K. M., Sidle, R. C. and Mallants, D.: Criteria for selecting fluorescent dye tracers for soil hydrological applications using Uranine as an example, J. Hydrol. Hydromechanics, 61(4), 313–325, doi:10.2478/johh-2013-0040, 2013.

Kristensen, E., Penha-Lopes, G., Delefosse, M., Valdemarsen, T., Quintana, C. O. and Banta, G. T.: What is bioturbation? the need for a precise definition for fauna in aquatic sciences, Mar. Ecol. Prog. Ser., 446, 285–302, doi:10.3354/meps09506, 2012.

Magni, P. and Montani, S.: Seasonal patterns of pore-water nutrients, benthic chlorophyll a and sedimentary AVS in a macrobenthos-rich tidal flat, Hydrobiologia, 571(1), 297–311, doi:10.1007/s10750-006-0242-9, 2006.

Maire, O., Merchant, J. N., Bulling, M., Teal, L. R., Grémare, A., Duchêne, J. C. and Solan, M.: Indirect effects of non-lethal predation on bivalve activity and sediment reworking, J. Exp. Mar. Bio. Ecol., 395(1–2), 30–36, doi:10.1016/j.jembe.2010.08.004, 2010.

Meysman, F. J. R., Galaktionov, O. S., Gribsholt, B. and Middelburg, J. J.: Bioirrigation in permeable sediments: Advective pore-water transport induced by burrow ventilation, Limnol. Oceanogr., 51(1), 142–156, doi:10.4319/lo.2006.51.1.0142, 2006.

Queirios, A. M., Stephens, N., Cook, R., Ravaglioli, C., Nunes, J., Dashfield, S., Harris, C., Tilstone, G. H., Fishwick, J., Braeckman, U., Somerfield, P. J. and Widdicombe, S.: Can benthic community structure be used to predict the process of bioturbation in real ecosystems?, Prog. Oceanogr., 137(April), 559–569, doi:10.1016/j.pocean.2015.04.027, 2015.

Ragueneau, O., Chauvaud, L., Moriceau, B., Leynaert, A., Thouzeau, G., Donval, A., Le Loc'h, F. and Jean, F.: Biodeposition by an invasive suspension feeder impacts the biogeochemical cycle of Si in a coastal ecosystem (Bay of Brest, France), Biogeochemistry, 75(1), 19–41, doi:10.1007/s10533-004-5677-3, 2005.

Rao, A. M. F., Malkin, S. Y., Montserrat, F. and Meysman, F. J. R.: Alkalinity production in intertidal sands intensified by lugworm bioirrigation, Estuar. Coast. Shelf Sci., 148, 36–47, doi:10.1016/j.ecss.2014.06.006, 2014.

Renz, J. R., Powilleit, M., Gogina, M., Zettler, M. L., Morys, C. and Forster, S.: Community bioirrigation potential ( BIP c ), an index to quantify the potential for solute exchange at the sediment-water interface, Mar. Environ. Res., (July), 0–1, doi:10.1016/j.marenvres.2018.09.013, 2018.

Roskosch, A., Hupfer, M., Nützmann, G. and Lewandowski, J.: Measurement techniques for quantification of pumping activity of invertebrates in small burrows, Fundam. Appl. Limnol. / Arch. für Hydrobiol., 178(2), 89–110, doi:10.1127/1863-9135/2011/0178-0089, 2011.

Schlüter, M., Sauter, E., Hansen, H. P. and Suess, E.: Seasonal variations of bioirrigation in coastal sediments: Modelling of field data, Geochim. Cosmochim. Acta, 64(5), 821–834, doi:10.1016/S0016-7037(99)00375-0, 2000.

De Smet, B., Braeckman, U., Soetaert, K., Vincx, M. and Vanaverbeke, J.: Predator effects on the feeding and bioirrigation activity of ecosystem-engineered Lanice conchilega reefs, J. Exp. Mar. Bio. Ecol., 475, 31–37, doi:10.1016/j.jembe.2015.11.005, 2016.

Stief, P. and de Beer, D.: Probing the microenvironment of freshwater sediment macrofauna: Implications of deposit-feeding and bioirrigation for nitrogen cycling, Limnol. Oceanogr., 51(6), 2538–2548, doi:10.4319/lo.2006.51.6.2538, 2006.

Wrede, A., Beermann, J., Dannheim, J., Gutow, L. and Brey, T.: Organism functional traits and ecosystem supporting services – A novel approach to predict bioirrigation, Ecol. Indic., 91, 737–743, doi:10.1016/j.ecolind.2018.04.026, 2018.

---

## Referee Comment (RC2) · Anonymous Referee #2 · 9 Dec 2019

This is a really nice study and a very interesting dataset. I have a few concerns but I think if these are addressed that this manuscript will be a really nice contribution to our understanding of bioirrigation.

Major concerns:

The manuscript starts off with a really nice, clearly written introduction to the importance of bioirrigation. However, the authors don't give a clear question or hypothesis at the end of the introduction, and the lead-in to the goal of the paper is a little confusing. To me, the question should be whether the bioirrigation potential calculated from community structure actually predicts the measured bioirrigation. I am somewhat skeptical

of this, and the data presented seems to indicate that the bioirrigation potential explains only a small amount of the variability in bioirrigation at most. The authors present this concept as if it is well established and a valid metric for characterizing bioirrigation, when it was introduced in 2018 and has not to my knowledge really been tested.

This lack of a clear question and testable hypothesis at the beginning leads to a number of other problems in the paper – the methods section is confusing because it consists of a list of methods without justifying exactly why the authors are doing these things. For example, there are measurements of chl a, C, N, grain size, but how do these relate to bioirrigation? What does it mean if these values are high or low? They clearly do relate to bioirrigation through sediment permeability, but this needs to be explicitly stated and justified. There was actually surprisingly little discussion of permeability, which is likely very important in bioirrigation.

More problematic, section 2.5 "Statistical analysis" was extremely difficult to follow, which made understanding and interpreting the results (Fig. 4) extremely difficult. These tests need to be tied to an explicit hypothesis and justified. Why were data on chl, grain size, etc., categorized rather than left linear? Why were absolute species abundances transformed to relative abundances? This doesn't make sense when considering the effect on bioirrigation, but it's not actually clear what this test was for. I understood very little of Fig. 4, it was confusing which datasets were used or what this figure is supposed to show, and what is the inset in fig. 4a?

The conclusions section was confusing and hard to follow since I didn't really follow the community analysis, need to revise around a clear question/hypothesis.

I was a little confused at times about why the authors were using abundance versus biomass – it seems to me that biomass is much more appropriate for predicting bioirrigation, so I don't understand, e.g., why they calculated an individual irrigation rate or what that is supposed to mean. Wouldn't the individual irrigation rate depend strongly on the size of the individual? It would make more sense to have an irrigation rate

normalized to the biomass, if normalization is useful.

Why were the data from subtidal and intertidal sites averaged? From Fig. 2, there are fairly big differences among sites, e.g., density among the intertidal sites, so this seems questionable.

I recommend re-framing the manuscript around the question of whether variability in community structure drives variability in bioirrigation or if other factors, e.g., season or temperature, sediment properties, subtidal vs intertidal, etc., are more important in driving bioirrigation. It's possible that this or something like this is what the community analysis was trying to get at, but it didn't come across clearly. Perhaps a generalized linear model predicting the measured r and a parameters from IPc and other variables would be more appropriate?

I think the high temporal resolution data on bioirrigation is really exciting and the analysis of those experiments is very interesting. While I haven't exhaustively read the literature on bioirrigation, these methods seem novel and exciting to me, and I encourage the authors to be a little more clear in taking credit for this, or at least be a little more explicit about whether this approach is new or how they built on previous studies. E.g., more detail on the models mentioned in line 46 and how this approach builds on those would be interesting to know. I really like the model presented in eqs 1-3, and encourage the authors to present more detail about the model results. For example, is it possible to separate the relative contribution of diffusion, advection, and adsorption (the 3 components in eq. 1) to the change in tracer concentration? In line 202, the authors mention that k (in the adsorption component) has much less impact than r and a, but I wasn't really sure how to interpret that. If I wanted to use this model to do a similar study, could I remove the adsorption term altogether or would I need to go to the trouble to measure k? Under what circumstances would it be okay to ignore that term? Similarly, diffusion is probably much smaller than advection – are there any circumstances in which diffusion is important?

I really like Fig. 3 and think that Fig. 3b does a great job of explaining the model output. It might be useful to show a few different examples in addition to the one in 3a to better illustrate the range of variability in bioirrigation among the samples. It seems to me that the attenuation coefficient reflects the volume of burrows rather than the depth of burrows. It's possible I'm not understanding this correctly, and certainly depth and volume would be correlated, but it makes more sense to me to think about volume rather than depth when thinking about dilution of tracer in the volume of overlying water.

Minor comments:

Line 12 – I disagree that using biological traits is "quantification" Line 42 – should explicitly state assumptions in the bioirrigation potential calculation Line 60 – was this calibration with bromide done using the linear fit or mechanistic model? Line 70 – were the two years averaged for the fall season? Interannual variability can be substantial. Line 77 – don't understand "70 samples" – is this pairs of cores or individual cores? Line 81 – should give data for average water temperatures, I suggest including this in the model of factors that affect irrigation Line 157 – Do you mean supplementary table 2? Line 241 – there is no Fig. 5 – do you mean 4? Discussion 4.1 – I don't understand the term "Bio-irrigation shape" – rephrase? Fig. 3 – it took me a minute to figure out that the y-axis in b was also ug L-1. Since the numbers are different, it would be useful to label this. I also suggest making the axis scales more similar - even though the starting points are different, the scale could be the same. Table 2 – suggest using "d50" instead of "MGS"

---

## Author Comment (AC2) · 7 Jan 2020

We thank the reviewer for insightful comments and suggestions, which improved the manuscript. The reviewer highlighted some confusing paragraphs in the materials and methods and discussion section, that relate to: the lack of a clear question/hypothesis statement in the introduction, and the use of the coinertia analysis as our multivariate analysis of choice.

These remarks were addressed in the following ways:
- Expansion of the introduction, by rephrasing and clarifying the aim of the study, and elaborating on the model development.
- Rewriting the materials and methods of the data analysis, and the caption of the main figure describing the coinertia analysis results.

Below we reply to the reviewer remarks, with the according line numbers where appropriate.

**1. Reviewer's comment:** *However, the authors don't give a clear question or hypothesis at the end of the introduction, and the lead-in to the goal of the paper is a little confusing. To me, the question should be whether the bioirrigation potential calculated from community structure actually predicts the measured bioirrigation. I am somewhat skeptical of this, and the data presented seems to indicate that the bioirrigation potential explains only a small amount of the variability in bioirrigation at most. The authors present this concept as if it is well established and a valid metric for characterizing bioirrigation, when it was introduced in 2018 and has not to my knowledge really been tested.*

**Reply:** By using the words "estimate"/"estimates" when describing the index approach, we tried to emphasize the difference between a quantitative measurement, and an ecological index application. Furthermore, in lines 56-62 we note the potential of this index, but also indicate that it is not really clear what this represents in terms of an ecological function. To clarify this, we edited line 43: "Bio-irrigation rates can be quantified with biogeochemical methods, or a qualitative estimate can be calculated by an index of bio-irrigation based on biological information."

We rephrased lines 63 – 68 to include an explicit statement of the aim of the study. "The aim of the current study was to compare bio-irrigation rate measurements with an index of bio-irrigation in natural sediments of a temperate estuarine system, the Oosterschelde. Samples were collected across different seasons in three subtidal and three intertidal sites with different benthic communities, and sediments varying from muddy to sandy. Bio-irrigation rates were derived by fitting a novel mechanistic model through a quasi-continuous time series of a fluorescent tracer, while biological information was used to calculate the IPc index."

**2. Reviewer's comment:** *the methods section is confusing because it consists of a list of methods without justifying exactly why the authors are doing these things. For example, there are measurements of chl a, C, N, grain size, but how do these relate to bioirrigation? What does it mean if these values are high or low? They clearly do relate to bioirrigation through sediment permeability, but this needs to be explicitly stated and justified. There was actually surprisingly little discussion of permeability, which is likely very important in bioirrigation.*

**Reply:** We added additional clarification to our choice of parameters by changing lines 78 - 80 to: "Sediment permeability has a strong influence on bio-irrigation rates (Aller, 1983; Meysman et al., 2006). Sediment permeability was not directly measured, but additional samples for sediment characteristics relating to this property (grain size distribution and porosity) were taken from the top 2 cm of sediment at each site, using a cut-off syringe. From the same samples a subsample was collected for determining the chlorophyll *a* content, and C/N ratios in the sediment, as measures of food availability and quality respectively."

Though we agree on the role of the sediment permeability for various functional properties, our results do not significantly contribute to the discussion about this topic in previous works (e.g. in: Aller, 1983; Meysman et al., 2006; Renz et al., 2018). The role of permeability in bio-irrigation is mentioned in the introduction (lines 36-40)**, it is included in the sampling design (through sediment grain size and porosity), and the efforts made by Renz et al. (2018) of including sediment type (~permeability) as a modifier in the index are mentioned in the discussion (lines 362-365).

**: We modified sentence 36-40 of the introduction to make the link with permeability stronger: "In muddy sediments, *where permeability is low*, bio-irrigation impacts are localized close to the burrow wall, as the transport of solutes radiating from the burrows is governed by diffusion (Aller, 1980)."

**3. Reviewer's comment:** *"Statistical analysis" was extremely difficult to follow, which made understanding and interpreting the results (Fig. 4) extremely difficult. These tests need to be tied to an explicit hypothesis and justified. Why were data on chl, grain size, etc., categorized rather than left linear? Why were absolute species abundances transformed to relative abundances? This doesn't make sense when considering the effect on bioirrigation, but it's not actually clear what this test was for.*

**Reply:** The materials and methods section "statistical analysis" was renamed "data analysis", and rewritten. The caption for figure 4 was also modified for clarity.

*Reasons to transform or categorize some variables*
Relative species abundances were used to emphasize the specific functional role of some species within the communities and to mask any size effect of total abundance that are often encountered on production gradients (Beauchard et al., 2017). The ordination techniques used here are based on linear relationships, hence they may fail in capturing possible non-linear relationships between some continuous variables. When variables are categorized, dummy variables (0 or 1) are used to represent the different categories of the original variable and this allows for the inclusion of non-linear relationships.

*Coinertia analysis*
A coinertia analysis basically combines ordinations of different datasets (e.g. environmental, biotic and biogeochemical) and tests whether these datasets are correlated.
With the coinertia analysis, and the following permutation test we first test the null hypothesis that there is no significant relationship between environmental variables and species densities. In our case, the null hypothesis is invalidated (second sentence of section 3.4, and figure 4a), this is visible as a clear co-structure between both datasets, hence the environmental variables are correlated with the species densities (and vice versa). We then couple the irrigation data to this co structure (visually, in Figure 4c), and test for correlation of these rates to the environment-species data (results in Table 5). With the coinertia analysis (Figure 4), the directionality of correlated variables also becomes immediately clear. The rewritten section, and the figure are included at the end of this rebuttal.

*Additional clarification*
In the results (3.4 Co-Inertia analysis), lines 222-226 were edited to include the result of the permutation test. This now reads: "The first and second axes of the co-inertia analysis (CoIA) explained 57% and 19% of the variance in the dataset respectively (histogram inset Fig. 4a). The Monte-Carlo permutation test resulted in a significant RV coefficient (the multivariate generalization of the squared Pearson correlation coefficient) of 0.62 (p < 0.001), showing that the species data and the environmental data are significantly correlated. Both the first and second axes of the MCA performed on the environmental parameters and of the PCA performed on the species community were correlated, indicated by high Pearson correlation coefficients (Figure 4a; for the first axis: r = 0.95, p < 0.001; for the second axis: r = 0.92, p < 0.001)."

**4. Reviewer's comment:** *I understood very little of Fig. 4, it was confusing which datasets were used or what this figure is supposed to show, and what is the inset in fig. 4a?*

**Reply:** By rewriting section 2.5, and the figure caption we hope to have clarified this. The inset in Figure 4a, displays the eigenvalue diagram of the co-structure resulting from the CoIA, and thus the relative importance of the first two axes (in black). This has been added to the figure caption.

**5. Reviewer's comment:** *The conclusions section was confusing and hard to follow since I didn't really follow the community analysis, need to revise around a clear question/hypothesis*

**Reply:** We rewrote the conclusion to : "By fitting a mechanistic model to fluorescent tracer measurements we were able to infer more detailed information on the bio-irrigation process in species communities than an exchange rate alone, thereby improving on linear regression techniques. Benthic organisms differ strongly in the magnitude and mode in which they express functional traits. With this study we aimed to determine whether bio-irrigation can be predicted by an index of bio-irrigation, calculated based on functional traits. This index was correlated to the attenuation coefficient, but not the bio-irrigation rate. Our findings also highlight the importance of the context in which indices for functional processes should be evaluated, because of the confounding role spatial context and behaviour play. Different species assemblages can have the same bio-irrigation rates, but differ in sediment depth over which they exchange solutes. This is important to consider when implementing bio-irrigation in models of sediment biogeochemistry." Also, by rewriting the explanation about the coinertia analysis (comment 4), we hope that the confusion has been removed.

**6. Reviewer's comment:** *I was a little confused at times about why the authors were using abundance versus biomass – it seems to me that biomass is much more appropriate for predicting bioirrigation, so I don't understand, e.g., why they calculated an individual irrigation rate or what that is supposed to mean. Wouldn't the individual irrigation rate depend strongly on the size of the individual? It would make more sense to have an irrigation rate normalized to the biomass, if normalization is useful.*

**Reply:** Large animals often dominate the assemblages in terms of biomass, even if they are present in very low numbers. These animals are often not adequately sampled with our sampling design. Even though they may not necessarily be active bio-irrigators they would dominate the analysis when based on biomass. In some of our samples, most of the biomass was in a large non-bio-irrigator (e.g. the furrow shell *Scrobicularia plana*). Analyses based on biomass strongly associated the measured bio-irrigation rate with the non-bio-irrigator, and diminish the importance of smaller bio-irrigating species in the same sample, which have a much smaller biomass (e.g. several *Hediste diversicolor*), but through their lifestyle are expected to contribute much more to bio-irrigation.

*Additional clarification*
Since the inclusion of the *irrdens* metric (irrigation divided by species density) is redundant for the discussion, we have decided to remove this metric and references to it from the manuscript. (In text on lines 234 to 237 and line 256; removed from tables 4 and 5; arrow removed from figure 4c).

**7. Reviewer's comment:** *Why were the data from subtidal and intertidal sites averaged? From Fig. 2, there are fairly big differences among sites, e.g., density among the intertidal sites, so this seems questionable*

**Reply:** Statistical tests were performed on the non-averaged data. To visualize the differences between the two habitats (intertidal and subtidal) in a table, we show average values with standard deviation as a measure for the variability between sites.

**8. Reviewer's comment:** *I recommend re-framing the manuscript around the question of whether variability in community structure drives variability in bioirrigation or if other factors, e.g., season or temperature, sediment properties, subtidal vs intertidal, etc., are more important in driving bioirrigation. It's possible that this or something like this is what the community analysis was trying to get at, but it didn't come across clearly. Perhaps a generalized linear model predicting the measured r and a parameters from IPc and other variables would be more appropriate?*

**Reply:** In complying with comment 1: "*To me, the question should be whether the bioirrigation potential calculated from community structure actually predicts the measured bioirrigation. I am somewhat skeptical of this, and the data presented seems to indicate that the bioirrigation potential explains only a small amount of the variability in bioirrigation at most.*", we did this by providing a clearer statement of the aims of the study (please see our reply to

comment 1, lines 63-68), which also includes seasonality, and the differences between habitats (subtidal – intertidal, sediment types) as possible influences on the bio-irrigation measurements, and index calculation.

A generalized linear model was part of our initial data-exploration process, but this did not give meaningful results. Therefore we chose to work with the coinertia approach, since this approach visualizes all the measured parameters, as well as the effect of dominant species, and still delivers *p*-statistics when combined with correlations.

**9. Reviewer's comment:** *I think the high temporal resolution data on bioirrigation is really exciting and the analysis of those experiments is very interesting. While I haven't exhaustively read the literature on bioirrigation, these methods seem novel and exciting to me, and I encourage the authors to be a little more clear in taking credit for this, or at least be a little more explicit about whether this approach is new or how they built on previous studies. E.g., more detail on the models mentioned in line 46 and how this approach builds on those would be interesting to know. I really like the model presented in eqs 1-3, and encourage the authors to present more detail about the model results. For example, is it possible to separate the relative contribution of diffusion, advection, and adsorption (the 3 components in eq. 1) to the change in tracer concentration?*

**Reply:** We added more detail on the models in lines 44-48. This paragraph now reads: "The biogeochemical methods estimate the exchange rates of a tracer substance (usually inert) between the overlying water and the sediment, by fitting a linear model (De Smet et al., 2016; Mestdagh et al., 2018; Wrede et al., 2018), or a quasi-mechanistic model (Berelson et al., 1998; Andersson et al., 2006) through measured concentration time series. A linear decrease returns the rate of disappearance of the tracer from the water column over a given time period, but it gives little information on the bio-irrigation process itself, e.g. what is the actual pumping rate, and where in the sediment are solutes exchanged. While sometimes the depth distribution of the tracer in the sediment is characterized post-experiment to obtain this information (Martin and Banta, 1992; Berg et al., 2001; Hedman et al., 2011), this step is often overlooked. By increasing the temporal resolution of the tracer concentration measurements, an exponential decrease can be fitted through the data, from which a bio-irrigation rate can be derived which is independent of the length of the experiment (Meysman et al., 2006; Na et al., 2008). For these applications fluorescent tracers are used, as they can be monitored in-situ, and the measurement is instantaneous. So far, this method has been applied in controlled settings, but not yet in field applications."

We believe that it would not be meaningful to separate the tracer exchange in terms of diffusion – advection and adsorption for a conservative tracer, as this would change significantly over time, and as the equilibrium solution is one in which the tracer is distributed evenly throughout the water and the sediment column, so the contribution of all these processes would be zero. However, it would be really worthwhile (albeit not within the scope of this study) to do this calculation for a non-conservative tracer.

To give more detail on model results, we have added more examples of fitted output (please see comment 11).

**10. Reviewer's comment:** *In line 202, the authors mention that k (in the adsorption component) has much less impact than r and a, but I wasn't really sure how to interpret that. If I wanted to use this model to do a similar study, could I remove the adsorption term altogether or would I need to go to the trouble to measure k? Under what circumstances would it be okay to ignore that term? Similarly, diffusion is probably much smaller than advection – are there any circumstances in which diffusion is important?*

**Reply:** The adsorption rate $k$ (hour$^{-1}$) has a small effect on the tracer profile because also the adsorption equilibrium term ($Eq_A$) is low. However, the importance of $k$ will increase with an increasing $Eq_A$, as both are multiplied. Although $Eq_A$ is low in our experiment, it is a significant parameter in the model, so both parameters should be measured, although it is experimentally challenging (Hameed and Ahmad, 2009). please see the supplement and reply to reviewer 1.

**11. Reviewer comment:** *I really like Fig. 3 and think that Fig. 3b does a great job of explaining the model output. It might be useful to show a few different examples in addition to the one in 3a to better illustrate the range of variability in bioirrigation among the samples.*

**Reply:** We have added additional examples, which were included in the supplement. The caption of figure 3 was modified to refer to these extra examples in the supplement.

[Figure]

**Figure R1: Additional model fits of data (red lines) collected different sites, at different times throughout the year. The best fit tracer profile (full black line) is shown, along with the range of model results as quantiles (light and dark grey).**

**12. Reviewer comment:** *It seems to me that the attenuation coefficient reflects the volume of burrows rather than the depth of burrows. It's possible I'm not understanding this correctly, and certainly depth and volume would be correlated, but it makes more sense to me to think about volume rather than depth when thinking about dilution of tracer in the volume of overlying water*

**Reply:** In our 1D model, the depth is indeed directly related to the volume, as it is assumed that the sediment is laterally homogenous (burrows are not explicitly modelled). The attenuation coefficient assigns to proportion of the total exchange that works on the different depth layers. In reality a high volume of burrows doesn't necessarily increase the depth (and decrease the attenuation coefficient). If all these burrows are shallow, the pumping rate will most likely increase, but the solute exchange could be concentrated in the first 3 cm for example. For increasing burrow densities of similar depths (and assuming no major species interaction effects), the attenuation will indeed decrease, as the exchange takes place equally over a larger depth range.

**13. Reviewer's comment:** Line 12 – *I disagree that using biological traits is "quantification".*

**Reply:** We agree with this comment, line 12 was changed to: "bio-irrigation is either quantified based on tracer data or, a community (bio-) irrigation potential (IPc) can be derived based on biological traits".

**14. Reviewer's comment:** *Line 42 – should explicitly state assumptions in the bioirrigation potential calculation.*

**Reply:** The assumptions were added to the manuscript on lines 51-...: "The index approach starts with the quantification of the abundance and biomass of organisms inhabiting the sediment, and an assessment of how these organisms bio-irrigate. The latter is done based on a set of life history traits which are assumed to contribute to bio-irrigation: the type of burrow they inhabit, their feeding type and their burrowing depth. Species are assigned one trait score for each trait, independent of the biological context in which they occur (but see Renz et al. (2018)). The species biomass and abundance, combined with their trait scores are then used to derive an index that represents the community (bio-) irrigation potential (BIPc and IPc in Renz et al., 2018 and Wrede et al., 2018 respectively), a similar practice to what is done for bioturbation with the community bioturbation potential (BPc; Queirós et al., 2013). The inherent assumptions of this approach are that bio-irrigation activity increases linearly with the number of organisms, and scales with their mean weight through a metabolic scaling factor."

**15. Reviewer's comment:** *Line 60 – was this calibration with bromide done using the linear fit or mechanistic model?*

**Reply:** This was done using the linear fit, we also cite this work in a previous line about this ". The biogeochemical methods estimate the exchange rates of a tracer substance (usually inert) between the overlying water and the sediment, by fitting a linear model (De Smet et al., 2016; Mestdagh et al., 2018; Wrede et al., 2018), or a quasi-mechanistic model (Berelson et al., 1998; Andersson et al., 2006) through measured concentration time series."

**16. Reviewer's comment:** *Line 70 – were the two years averaged for the fall season? Interannual variability can be substantial.*

**Reply:** These were averaged, as the range in the data was not out of the ordinary for either the species data (Figure 2), or the environmental data (Table 4).

**17. Reviewer's comment:** *Line 77 – don't understand "70 samples" – is this pairs of cores or individual cores?*

**Reply:** To clarify we changed it to the following: "In total 70 individual cores in the intertidal, and 47 in the subtidal were collected."

**18. Reviewer's comment:** *Line 81 – should give data for average water temperatures, I suggest including this in the model of factors that affect irrigation*

**Reply:** Average temperatures of the water in the cores during measurements were added to table 1, and referred to in methods on line 81. The effect of temperature is implicitly included in the factor season, used in the multivariate analysis.

| Season | Spring | Summer | Autumn | Winter |
|---|---|---|---|---|
| Months | Apr – Jun | Jul – Sep | Oct – Dec | Jan – Mar |
| Avg. Temperature (°C) | 12.8 | 17.9 | 11.9 | 7.3 |
| Dortsman | 4 | 5 | 9 | 5 |
| Zandkreek | 4 | 6 | 9 | 6 |
| Olzendenpoder | 4 | 4 | 8 | 6 |
| Lodijksegat | 4 | 4 | 8 | 2 |
| Hammen | 4 | 4 | 8 | 2 |
| Viane | 3 | 0 | 6 | 2 |

**19. Reviewer's comment:** *Line 157 – Do you mean supplementary table 2?*

**Reply:** Thank you for noticing this, it should be Figure 2 of the manuscript and has been corrected as such.

**20. Reviewer's comment:** *Line 241 – there is no Fig. 5 – do you mean 4?*

**Reply:** Thank you for noticing this, it has been corrected.

**21. Reviewer's comment:** *Discussion 4.1 – I don't understand the term "Bio-irrigation shape" – rephrase?*

**Reply:** We changed this title to "Advantages of mechanistic modelling", as this conveys the information in the following paragraphs more clearly.

**22. Reviewer's comment:** *Fig. 3 – it took me a minute to figure out that the y-axis in b was also ug L-1. Since the numbers are different, it would be useful to label this. I also suggest making the axis scales more similar – even though the starting points are different, the scale could be the same.*

**Reply:** We added the units to the y-axis in the b-part. The scale has not been changed because then the inset does not fit on the figure anymore. If we change the scale on the a part, then this generates a large blank space in the domain as well. The purpose of this figure is also not to compare a and b, they both have their individual explanation.

**23. Reviewer's comment:** *Table 2 – suggest using "d50" instead of "MGS"*

**Reply:** We changed it both here, and in section 3.1 of the results.

**REWRITTEN SECTION**

**2.5 Data analysis**

Differences in model derived pumping rates r and attenuation coefficient a between subtidal and intertidal were tested using a two-sided T-test (using a significance level of 0.05). For further multivariate analysis, species densities, biomass, and estimated irrigation parameters were averaged per station, and per season (Figure 2) since not all six stations were sampled on the same date. The patterns in abiotic conditions, species composition and bio-irrigation rates were analysed using ordination techniques for multivariate datasets as described in Thioulouse et al.(2018), and implemented in the ade4 R package (Dray and Dufour, 2015). In this procedure, a coinertia analysis and permutation first tests the null hypothesis that there is no significant relationship between environmental variables and species densities, and then the correlation of the bio-irrigation rates to the environment-species data is assessed. In a first step, the species data matrix was processed by centered Principle Component Analysis (PCA). For this the species relative densities were used to emphasize the specific functional role of some species within the communities (Beauchard et al., 2017). Secondly the environmental variable matrix

was processed by Multiple Correspondence Analysis (MCA; Tenenhaus and Young (1985). This technique can account for non-linear relationships between variables, but requires all variables to be categorical. Sediments were categorized based on grain size into the Udden-Wentworth scale (Wentworth, 1922) of silt (< 63 μm), very fine sand (> 63 μm, < 125 μm) and fine sand (> 125 μm, < 250 μm); the Chl a content was categorized to distinguish sites with low (< 8 μg g-1), intermediate (8-16 μg g-1) and high (> 16 μg g-1) chlorophyll content. Two abiotic variables were already categorical: habitat type (intertidal versus subtidal) and season. Sediment porosity and C/N ratio were not used in the analysis given the small range within these data (Table 2). In a third step, the two ordinations were combined in a Co-Inertia Analysis (CoIA; Dray et al. (2003)), to explore the co-structure between the species and the environmental variables. The significance of the overall relationship (the co-structure of species and environment) between the two matrices was tested by a Monte-Carlo procedure based on 999 random permutations of the row matrices (Heo and Gabriel, 1998). Finally, the correlations between the response variables relating to irrigation (estimated irrigation parameters, calculated IPc, BPc) and the two axes of the co-inertia analysis were assessed using the Pearson correlation coefficient assuming a significance level of 0.05. Results are expressed as mean ± sd.

[Figure]

**Figure 4: Summary of the coinertia analysis (CoIA). (a) Co-structure between abiotic samples (circles) and species samples (arrow tips); grey circles "D", "O","Z" for intertidal sites Dortsman, Olzendenpolder and Zandkreek respectively; white circles "H", "L", "V" for subtidal sites Hammen, Lodijksegat and Viane respectively. Arrow length corresponds to the dissimilarity between the abiotic data and the species data (the larger the arrow, the larger the dissimilarity). Pearson's correlation between the circle and arrow tip coordinates on the first axis: r = 0.95, p < 0.001; on the second axis, r = 0.92, p < 0.001. Sites are more similar in terms of environmental conditions (circles), or species (arrow tips), when they group closer together. Inset: eigenvalue diagram of the co-structure; first axis explains 57%, second axis explains 19% of the variation in the dataset. (b) MBA based on environmental variables. (c) Species projections (dark arrows) and projected response variables (bio-irrigation parameters and bioturbation and bio-irrigation index) onto the co-inertia axes (grey arrows). The directions of arrows in figures b and c corresponds to the directions in which stations are grouped in terms of abiotic data (circles) and species composition (arrow tips) in figure a.**

**References**

Aller, R. C.: The importance of the diffusive permeability of animal burrow linings in determining marine sediment chemistry., J. Mar. Res., 41(2), 299–322, doi:10.1357/002224083788520225, 1983.

Beauchard, O., Veríssimo, H., Queirós, A. M. and Herman, P. M. J.: The use of multiple biological traits in marine community ecology and its potential in ecological indicator development, Ecol. Indic., 76, 81–96,

doi:10.1016/j.ecolind.2017.01.011, 2017.

Dray, S. and Dufour, A.-B.: The ade4 Package: Implementing the Duality Diagram for Ecologists, J. Stat. Softw., 22(4), doi:10.18637/jss.v022.i04, 2015.

Dray, S., Chessel, D. and Thioulouse, J.: Co-inertia analysis and the linking of ecological data tables, Ecology, 84(11), 3078–3089, doi:10.1890/03-0178, 2003.

Hameed, B. H. and Ahmad, A. A.: Batch adsorption of methylene blue from aqueous solution by garlic peel, an agricultural waste biomass, J. Hazard. Mater., 164(2–3), 870–875, doi:10.1016/j.jhazmat.2008.08.084, 2009.

Heo, M. and Gabriel, K. R.: A permutation test of association between configurations by means of the RV coefficient, Commun. Stat. Part B Simul. Comput., 27(3), 843–856, doi:10.1080/03610919808813512, 1998.

Meysman, F. J. R., Galaktionov, O. S., Gribsholt, B. and Middelburg, J. J.: Bio-irrigation in permeable sediments: An assessment of model complexity, J. Mar. Res., 64(4), 589–627, doi:10.1357/002224006778715757, 2006a.

Meysman, F. J. R., Galaktionov, O. S., Gribsholt, B. and Middelburg, J. J.: Bioirrigation in permeable sediments: Advective pore-water transport induced by burrow ventilation, Limnol. Oceanogr., 51(1), 142–156, doi:10.4319/lo.2006.51.1.0142, 2006b.

Na, T., Gribsholt, B., Galaktionov, O. S., Lee, T. and Meysman, F. J. R.: Influence of advective bio-irrigation on carbon and nitrogen cycling in sandy sediments, J. Mar. Res., 66, 691–722, doi:10.1357/002224008787536826, 2008.

Queirós, A. M., Birchenough, S. N. R., Bremner, J., Godbold, J. A., Parker, R. E., Romero-Ramirez, A., Reiss, H., Solan, M., Somerfield, P. J., Van Colen, C., Van Hoey, G. and Widdicombe, S.: A bioturbation classification of European marine infaunal invertebrates, Ecol. Evol., 3(11), 3958–3985, doi:10.1002/ece3.769, 2013.

Renz, J. R., Powilleit, M., Gogina, M., Zettler, M. L., Morys, C. and Forster, S.: Community bioirrigation potential ( BIP c ), an index to quantify the potential for solute exchange at the sediment-water interface, Mar. Environ. Res., (July), 0–1, doi:10.1016/j.marenvres.2018.09.013, 2018.

Tenenhaus, M. and Young, F. W.: An analysis and synthesis of multiple correspondence analysis, optimal scaling, dual scaling, homogeneity analysis and other methods for quantifying categorical multivariate data, Psychometrika, 50(1), 91–119, doi:10.1007/BF02294151, 1985.

Thioulouse, J., Dray, S., Dufour, A.-B., Siberchicot, A., Jombart, T. and Pavoine, S.: Multivariate Analysis of Ecological Data, 1st ed., Springer-Verlag New York, New York., 2018.

Wrede, A., Beermann, J., Dannheim, J., Gutow, L. and Brey, T.: Organism functional traits and ecosystem supporting services – A novel approach to predict bioirrigation, Ecol. Indic., 91(April), 737–743, doi:10.1016/j.ecolind.2018.04.026, 2018.